# Conditional chemoconnectomics (cCCTomics) as a strategy for efficient and conditional targeting of chemical transmission

**Renbo Mao**[1,2,3,4,5†], **Jianjun Yu**[1,2,3,4†], **Bowen Deng**[1,2,3,4], **Xihuimin Dai**[1,2,3,4], **Yuyao Du**[1,2,3,4], **Sujie Du**[1,2,3,4], **Wenxia Zhang**[1,2,3,4], **Yi Rao**[1,2,3,4]*

[1]Laboratory of Neurochemical Biology, Chinese Institute for Brain Research, Beijing, China; [2]PKU-IDG/McGovern Institute for Brain Research, Peking-Tsinghua Center for Life Sciences, School of Life Sciences, Department of Chemical Biology, College of Chemistry and Chemical Engineering, School of Pharmaceutical Sciences, Peking University, Beijing, China; [3]Chinese Institutes for Medical Research, Capital Medical University; Changping Laboratory, Changping, China; [4]Research Unit of Medical Neurobiology, Chinese Academy of Medical Sciences, Beijing, China; [5]National Institute of Biological Sciences, Chinese Academy of Medical Sciences & Peking Union Medical College, Beijing, China

**\*For correspondence:** yrao@pku.edu.cn

[†]Co-first authors

**Competing interest:** The authors declare that no competing interests exist.

**Sent for Review** 09 September 2023
**Preprint posted** 27 September 2023
**Reviewed preprint posted** 19 December 2023
**Reviewed preprint revised** 08 April 2024
**Version of Record published** 30 April 2024

**Abstract** Dissection of neural circuitry underlying behaviors is a central theme in neurobiology. We have previously proposed the concept of chemoconnectome (CCT) to cover the entire chemical transmission between neurons and target cells in an organism and created tools for studying it (CCTomics) by targeting all genes related to the CCT in *Drosophila*. Here we have created lines targeting the CCT in a conditional manner after modifying GFP RNA interference, Flp-out, and CRISPR/Cas9 technologies. All three strategies have been validated to be highly effective, with the best using chromatin-peptide fused Cas9 variants and scaffold optimized sgRNAs. As a proof of principle, we conducted a comprehensive intersection analysis of CCT genes expression profiles in the clock neurons, uncovering 43 CCT genes present in clock neurons. Specific elimination of each from clock neurons revealed that loss of the neuropeptide CNMa in two posterior dorsal clock neurons (DN1ps) or its receptor (CNMaR) caused advanced morning activity, indicating a suppressive role of CNMa-CNMaR on morning anticipation, opposite to the promoting role of PDF-PDFR on morning anticipation. These results demonstrate the effectiveness of conditional CCTomics and its tools created here and establish an antagonistic relationship between CNMa-CNMaR and PDF-PDFR signaling in regulating morning anticipation.

## eLife assessment

This article expands the genetic toolset that was previously developed by the Rao Lab to introduce the conditional downregulation of neurotransmission components in *Drosophila*. As a proof of principle, the authors tested their new collection and provide evidence of the contribution of CNMa-mide (a neuropeptide) to the temporal control of locomotor activity patterns. These are overall **important** findings supported by **compelling** evidence.

## Introduction

Much research efforts have been made to uncover the wiring and signaling pathways of neural circuits underlying specific behaviors. Circuit dissection strategies include genetic screening, genetic labeling, circuit tracing, live imaging, genetic sensors, and central nervous system (CNS) reconstruction via electron microscopy (EM). Recently, we have developed chemoconnectomics (CCTomics), focusing on building a comprehensive set of knockout and knockin tool lines of chemoconnectome (CCT) genes, to dissect neural circuitry based on chemical transmission (*Deng et al., 2019*).

Each strategy has advantages and disadvantages. For example, genetic screening is less biased but inefficient; circuit tracing with viruses provides information of connection, but is often prone to leaky expression and inaccurate labeling; and EM reconstruction is anatomically accurate but does not allow for manipulation of corresponding neurons. CCTomics overcomes limitations of previous strategies by allowing for behavioral screening of CCT genes and accurate labeling or manipulation of corresponding neurons. However, it is still limited in that knockout of some CCT genes can be lethal during development and that CCT genes may function differently in different neurons, which require a cell-type-specific manipulation. Thus, we decided to invent a conditional CCTomics (cCCTomics) in which gene deletion was conditional.

There are three major strategies for somatic gene mutagenesis at the DNA/RNA level: RNA interference, DNA site-specific recombination enzymes, and CRISPR/Cas system. RNA interference targets RNAs conveniently and efficiently (*Martin and Caplen, 2007*; *Oberdoerffer et al., 2005*). Libraries of transgenic RNAi flies covering almost the entire fly genome have been established (*Ni et al., 2011*; *Perkins et al., 2015*). DNA site-specific recombination enzymes such as Flp, B3, and Cre mediate specific and efficient gene editing (*Gaj et al., 2014*; *Grindley et al., 2006*). These strategies require flies with reverse repetitive sequences knocked into the corresponding genes, which is time-consuming with relatively complex recombination for genetic assays. CRISPR/Cas systems, particularly CRISPR/Cas9, which targets DNA with a sgRNA/Cas protein complex, have been broadly applied in gene manipulation over the last decade. The widespread use of CRIPSR/Cas9 in *Drosophila* somatic gene manipulation began in 2014 (*Xue et al., 2014*). Later, tRNA-flanking sgRNAs was designed and applied, which enabled multiple sgRNAs to mature in a single transcript (*Xie et al., 2015*), accelerating the application of this strategy in conditional gene manipulation in flies with impressive efficiency (*Delventhal et al., 2019*; *Port and Bullock, 2016*; *Port et al., 2020*; *Schlichting et al., 2019*; *Schlichting et al., 2022*). Additionally, libraries of UAS-sgRNA targeting kinases (*Port et al., 2020*) and GPCRs *Schlichting et al., 2022* have been established, but no sgRNA libraries covering all the CCT genes exist yet. The efficiency of CRISPR/Cas9 has not been validated systematically in the *Drosophila* nervous system.

The circadian rhythm can be used for proof-of-principle testing of cCCTomics. Organisms evolve periodic behaviors and physiological traits in response to cyclical environmental changes. The rhythmic locomotor behavior of *Drosophila*, for instance, shows enhanced activity before the light is turned on and off in a light–dark (LD) cycle, referred to as morning and evening anticipations, respectively (*Collins et al., 2005*; *Helfrich Förster, 2001*). Under 12 hr dark–12 hr dark (DD) conditions, the activities peak regularly about every 24 hr (*Konopka and Benzer, 1971*). Approximately 150 clock neurons, circadian output neurons, and extra-clock electrical oscillators (xCEOs) coordinate *Drosophila* circadian behaviors (*Dubowy and Sehgal, 2017*; *Tang et al., 2022*). The regulation of morning and evening anticipations, the most prominent features in the LD condition, is primarily mediated by four pairs of sLNvs expressing pigment dispersing factor (PDF), six pairs of LNds and the fifth s-LNv (*Grima et al., 2004*; *Rieger et al., 2006*; *Stoleru et al., 2004*). At the molecular level, Pdf and Pdf receptor (PDFR) are well known, with their mutants showing an advanced evening activity peak and no morning anticipation (*Hyun et al., 2005*; *Lear et al., 2005*; *Renn et al., 1999*). Other neuropeptides and their receptors, including AstC/AstC-R2 and neuropeptide F (NPF) and its receptor (NPFR), have also been reported to modulate evening activities (*Díaz et al., 2019*; *He et al., 2013*; *Hermann et al., 2012*), while CCHa1/CCHa1-R and Dh31 regulate morning activities (*Fujiwara et al., 2018*; *Goda et al., 2019*). To date, no advanced morning activity phenotype has been reported in flies.

To develop a more efficient approach for somatic gene manipulation, we have now generated two systems for conditional manipulation of CCT genes: (1) GFPi/Flp-out-based conditional knockout (cKO) system of CCT genes (cCCTomics) and (2) CRISPR/Cas9-based (C-cCCTomics). Both systems have achieved high efficiency of gene mutagenesis in the *Drosophila* nervous system. C-cCCTomics,

utilizing chromatin-peptide fused Cas9 and scaffold optimized sgRNA, makes efficient conditional gene knockout as simple as RNAi. Further application of C-cCCTomics in clock neurons revealed novel roles of CCT genes in circadian behavior: CNMa-CNMaR modulates morning anticipation as an antagonistic signal of PDF-PDFR.

## Results

### Near-complete disruption of target genes by GFPi and Flp-out-based cCCTomics

For the purpose of cCCTomics, we initially leveraged the benefits of our previously generated CCTomics attP lines (*Deng et al., 2019*), which enabled us to fuse enhanced GFP (EGFP) coding sequence at the 3′ end of each gene's coding region and flank most or entire gene span with FRT sequence through site-specific recombination (*Figure 1A* and *Supplementary file 1*). We designed this system so that it could be used to target genes tagged with GFP by RNAi (*Neumüller et al., 2012*) and at the same time to enable flippase (Flp)-mediated DNA fragment excision between two FRT sequences when FRT sequences are in the same orientation (*Vetter et al., 1983*; *Figure 1A*).

To validate the efficiency of cCCTomics, we performed pan-neuronal expression of either shRNA$^{GFP}$ or flipase in cCCT flies. Immunofluorescent imaging showed that constitutive expression of shRNA$^{GFP}$ (*Figure 1B–D*, *Figure 1—figure supplement 1A–I*) or flipase (*Figure 1E–G*, *Figure 1—figure supplement 1J–U*) almost completely eliminated GFP signals of target genes, indicating high efficiency. Knocking out at the adult stage using either hsFLP driven Flp-out (*Golic and Lindquist, 1989*; *Figure 1H–J*) or neural (elav-Switch) driven shRNA$^{GFP}$ (*Nicholson et al., 2008*; *Osterwalder et al., 2001*; *Figure 1—figure supplement 2A–I*) also resulted in the elimination of most, though not all, GFP signals. Notably, control group of CCT$^{EGFP.FRT}$; elav-Switch/UAS-shRNA$^{GFP}$ flies fed with solvent (ethanol) showed obvious decreased GFP (*Figure 1—figure supplement 2B, E, and H*) compared with UAS-shRNA$^{GFP}$/CCT$^{EGFP.FRT}$ flies fed with RU486 (*Figure 1—figure supplement 2A, D, and G*), indicating leaky expression of elav-Switch or shRNA$^{GFP}$.

We then applied cCCTomics in pan-neuronal knockout of nAChRβ2, which is required for *Drosophila* sleep (*Dai et al., 2021*). Ablation of nAChRβ2 in the nervous system dramatically decreased sleep of flies, mirroring the nAChRβ2 knockout phenotype (*Figure 1K and L*). Therefore, cCCTomics is an effective toolkit for manipulation of CCT genes and suitable for functional investigations of genes. Expression of in-frame fused EGFP-labeled CCT genes highly co-localized with signals revealed by immunocytochemistry (*Figure 1—figure supplement 3A–I*), allowing direct examination of gene expression without amplification, which is different from the GAL4/UAS binary system.

We then checked the viability of cCCT lines and found that cCCT lines including Capa$^{EGFP.FRT}$, ChAT$^{EGFP.FRT}$, and Eh$^{EGFP.FRT}$ were viable, whereas their CCT mutants were lethal. Gad1$^{EGFP.FRT}$, GluRI-ID$^{EGFP.FRT}$, and CapaR$^{EGFP.FRT}$ were still lethal as their CCT mutants were (*Supplementary file 2*). This indicates that some of the cCCT knockin flies may functionally affect corresponding genes, which are not suitable for conditional gene manipulation. Combination of cCCT transgenic flies with UAS-Flp, UAS-shRNA$^{GFP}$, or specific drivers is complicated and unfriendly for screen work despite the almost 100% efficiency of gene suppression when driven by a pan-neuronal driver. Because of the limitations of this method, we further created a CRISPR/Cas9-based cKO system of chemoconnectomics (C-cCCTomics).

### CRISPR/Cas9-based cKO system for CCTomics

To simplify effective manipulation of CCT genes, we designed a vector based on pACU2 (*Han et al., 2011*) with tRNA flanking sgRNAs (*Port and Bullock, 2016*; *Xie et al., 2015*) targeting CCT genes (*Figure 2A*). We also adopted an optimized sgRNA scaffold 'E+F' (E, stem extension; F, A-U flip) (*Chen et al., 2013*), which facilitates Cas9-sgRNA complex formation and gene knockout efficiency (*Dang et al., 2015*; *Poe et al., 2019*; *Zhao et al., 2016*), to all sgRNAs to improve gene knockout efficiency. To balance efficiency and off-target effect, we selected three sgRNAs for each CCT gene with the highest predicted efficiency and no predicted off-target effect based on previously reported models (*Chu et al., 2016*; *Doench et al., 2014*; *Graf et al., 2019*; *Gratz et al., 2014*; *Heigwer et al., 2014*; *Stemmer et al., 2015*; *Xu et al., 2015*; see *Supplementary file 3* and 'Materials and methods').

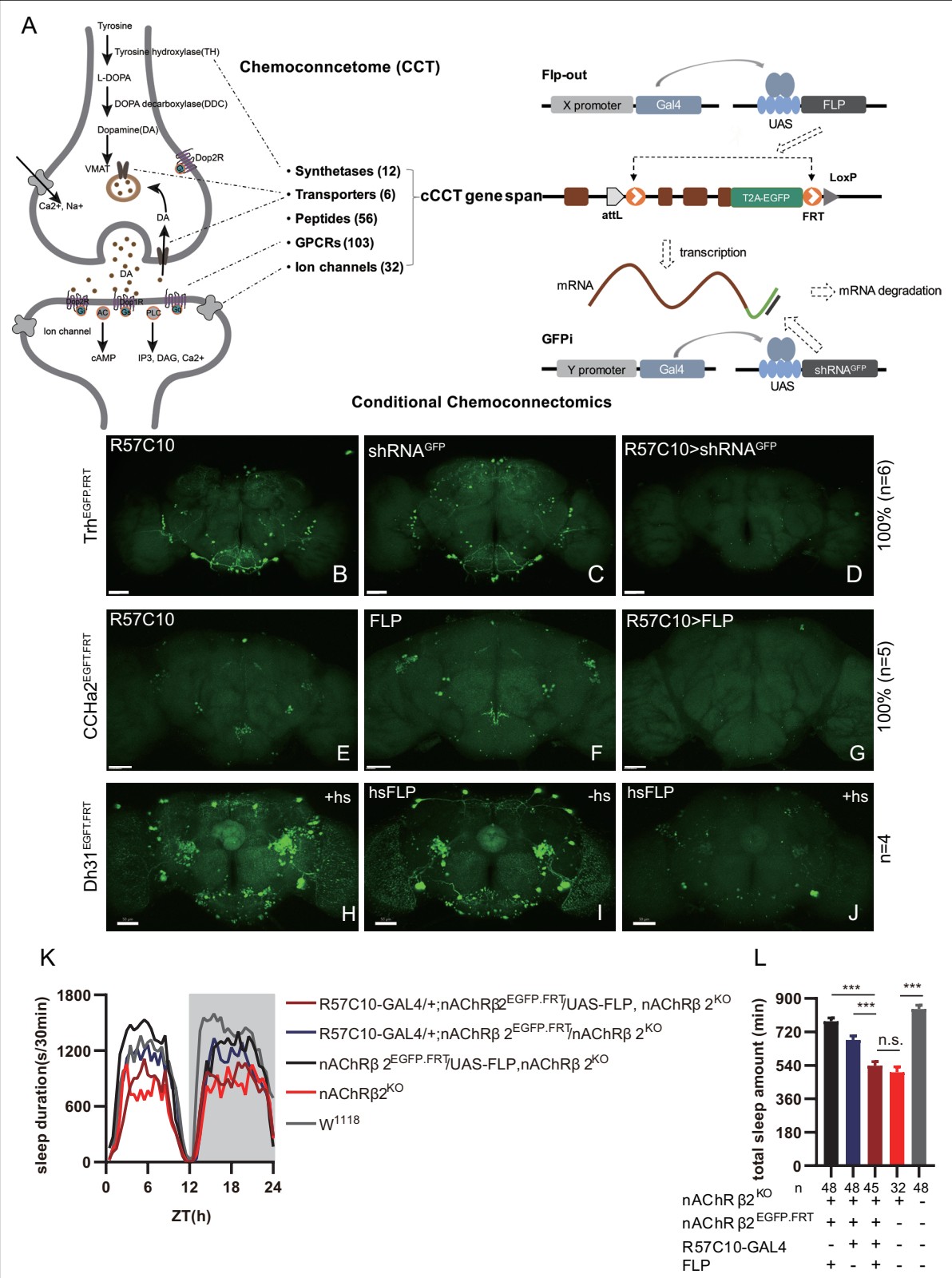

**Figure 1.** Conditional chemoconnectomics (cCCTomics) mediates efficient conditional disruption of chemoconnectome (CCT) genes. (**A**) Schematic of cCCT gene span and principle of cCCTomics. A T2A-EGFP sequence was introduced at the 3' end of CCT genes and their most or all coding regions (depending on attP-KO lines) were flanked by 34 bp FRT sequence. Both Flp-out (top) and GFP RNAi (down) could mediate CCT gene manipulation. (**B–J**) Expression of Trh (**B–D**), CCHa2 (**E–G**), and Dh31 (**H–J**) is efficiently disrupted by pan-neuronal expression of GFP-RNAi (**B–D**), pan-neuronal

*Figure 1 continued on next page*

*Figure 1 continued*

expression of Flp-out (**E–G**), and heatshock-Flp (**H–J**), respectively. Representative fluorescence images of R57C10-Gal4/+;Trh[EGFP.FRT]/+ (**B**), UAS-shRNA[GFP]/Trh[EGFP.FRT] (**C**), R57C10-Gal4/+; UAS-shRNA[GFP]/Trh[EGFP.FRT] (**D**), R57C10-Gal4/+;CCHa2[EGFP.FRT]/+; (**E**), UAS-Flp/CCHa2[EGFP.FRT] (**F**), R57C10-Gal4/+; UAS-Flp/CCHa2[EGFP.FRT] (**G**), Dh31[EGFP.FRT] with heatshock (**H**), hs-Flp/Dh31[EGFP.FRT] without heatshock (**I**), and hs-Flp/Dh31[EGFP.FRT] with heatshock are shown. Manipulation efficiency and experiment group fly number is noted on the right. Scale bar, 50 um. (**K, L**) sleep profiles (**K**) and statistical analysis (**L**) of Flp-out-induced nAChRβ2 neuronal knockout flies (dark red), nAChRβ2 knockout flies (light red), and genotype controls (dark, gray and blue). Sleep profiles are plotted in 30 min bins. In this and other figures, blank background indicates the light phase (ZT 0–12); shaded background indicates the dark phase (ZT 12–24). Daily sleep duration was significantly reduced in nAChRβ2 neuronal knockout files, which is comparable to nAChRβ2 knockout. In all statistical panels, unless otherwise noted, (1) numbers below each bar represent the number of flies tested. (2) Mean ± SEM is shown. (3) The Kruskal–Wallis test followed by Dunn's post test was used. \*\*\*p<0.001; \*\*p<0.01; \*p<0.05; n.s., p>0.05. Male flies were used unless otherwise noted.

The online version of this article includes the following source data and figure supplement(s) for figure 1:

**Source data 1.** Data points for *Figure 1K and L*.

**Figure supplement 1.** Efficient conditional disruption of chemoconnectome (CCT) genes by conditional chemoconnectomics (cCCTomics).

**Figure supplement 2.** Gene disruption of target genes by induced shRNA.

**Figure supplement 3.** Accurate labeling of target genes by cCCT lines.

We generated UAS-sgRNA[3x] transgenic lines for all 209 defined CCT genes (*Abruzzi et al., 2017*; *Dai et al., 2019*; *Deng et al., 2019*) and UAS-Cas9.HC (Cas9.HC) (*Mali et al., 2013*). We first verified that C-cCCTomics mediated precise target DNA breaking by ubiquitous expression of Cas9.P2 (*Port et al., 2014*) and sgRNA by targeting Pdf or Dh31. Sanger sequencing showed that indels were present exactly at the Cas9 cleavage sites (*Figure 2—figure supplement 1A–H*).

To determine the efficiency of C-cCCTomics, we employed pan-neuronal expression of Cas9.HC with sgRNA[Dh31] or sgRNA[pdf]. Targeting by Cas9.HC/sgRNA[Dh31] eliminated most but not all of the GFP signal in Dh31[EGFP.FRT] (*Figure 2B–D*), whereas all anti-PDF signals were eliminated by Cas9.HC/sgRNA[pdf] (*Figure 2E–G*). Furthermore, we used the C-cCCT strategy to conditionally knockout genes for Pdfr, nAChRβ2, and nAChRα2, which were previously reported as essential for circadian rhythm or sleep (*Dai et al., 2021*; *Renn et al., 1999*). Pan-neuronal knockout of Pdfr resulted in a tendency toward advanced evening activity and weaker morning anticipation compared to control flies (*Figure 2H and I*), which is similar to Pdfr-attpKO flies. These phenotypes were not as pronounced as those reported previously when han[5304] mutants exhibited a more obvious advanced evening peak and no morning anticipation (*Hyun et al., 2005*). Furthermore, there was no significant sleep decrease in these cKO flies (*Figure 2J and K*) when we applied C-cCCTomics to manipulate nAChRβ2 or nAChRa2. Taken together, C-cCCTomics (with Cas9.HC) achieved a relatively high gene knockout efficiency, but it was not effective enough for all genes.

## Evaluation of Cas9 with different chromatin-modulating peptides

Since the establishment of the CRISPR/Cas9 system a decade ago, many groups have attempted to improve its efficiency in gene manipulation. Most attempts have been focused on the two main components of this system, the Cas9 protein (*Ding et al., 2019*; *Ling et al., 2020*; *Liu et al., 2019*; *Zhao et al., 2016*; *Zheng et al., 2020*) and the single-guide RNA (*Chen et al., 2013*; *Chu et al., 2016*; *Dang et al., 2015*; *Doench et al., 2014*; *Filippova et al., 2019*; *Graf et al., 2019*; *Labuhn et al., 2018*; *Mu et al., 2019*; *Nahar et al., 2018*; *Scott et al., 2019*; *Xu et al., 2015*). At the beginning of C-cCCTomics design, we adopted an optimized sgRNA scaffold and selected sgRNAs with predicted high efficiency. We tried to further improve the efficacy by modifying Cas9 protein. We fused a chromatin-modulating peptide (*Ding et al., 2019*), HMGN1 (high mobility group nucleosome binding domain 1), at the N-terminus of Cas9 and HMGB1 (high mobility group protein B1) at its C-terminus with GGSGP linker, termed Cas9.M9 (*Figure 3A*, 'Materials and methods'). We also obtained a modified Cas9.M6 with HMGN1 at the N-terminus and an undefined peptide (UDP) at the C-terminus (*Figure 3A*). We replaced the STARD linker between Cas9 and NLS in Cas9.HC with the GGSGP linker (*Zhao et al., 2016*), termed Cas9.M0 (*Figure 3A*). None of these modifications have been validated previously in flies.

To determine whether the modified Cas9 variants were more efficient, we first pan-neuronally expressed each Cas9 variant and sgRNA[ple], and assessed their efficiency by immunofluorescence imaging. By counting anti-TH-positive neurons in the brain (anterior view) after targeting by Cas9/sgRNA[ple], we found that unmodified Cas9.HC/sgRNA[ple] only achieved 69.58 ± 3.04% (n = 5) knockout

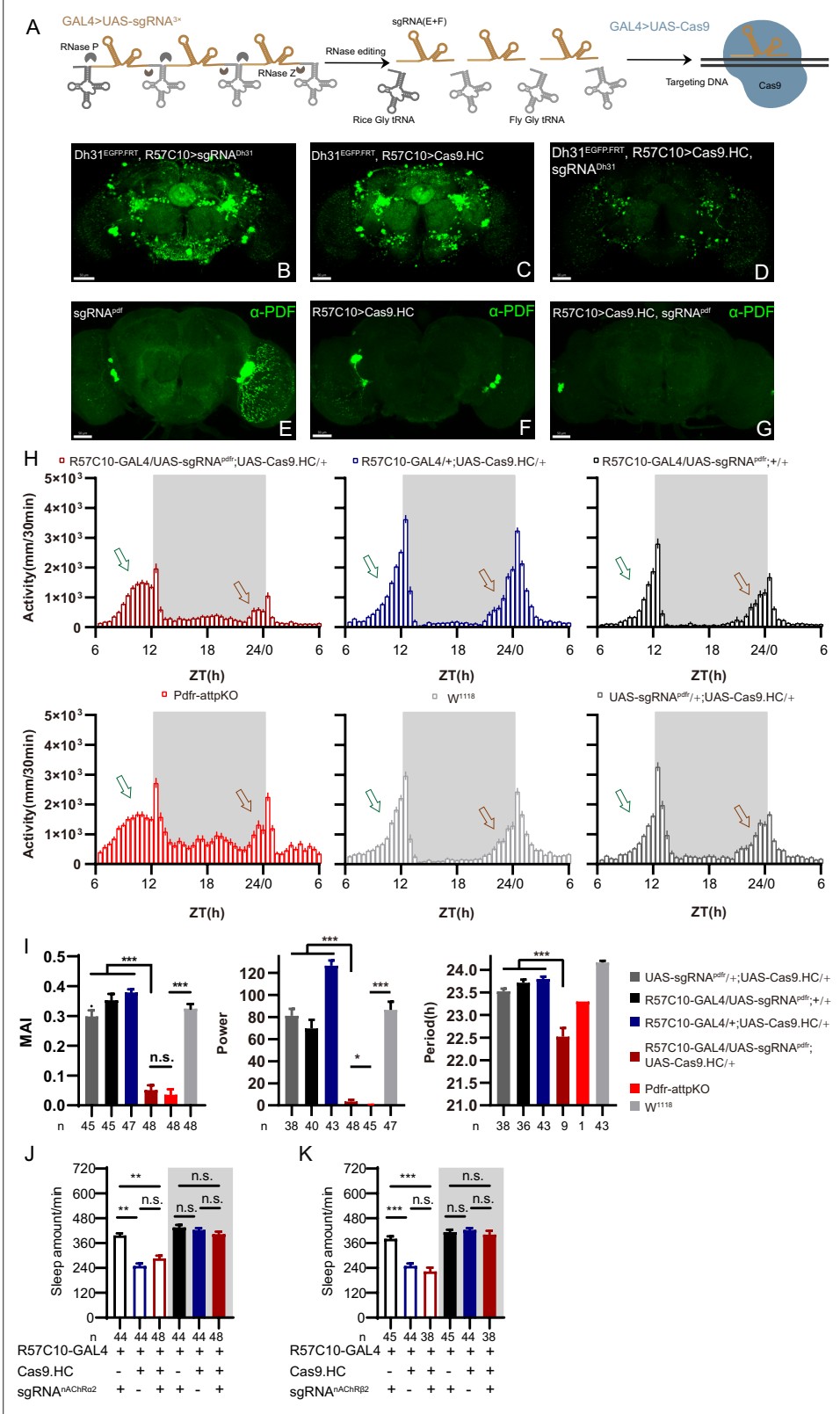

**Figure 2.** C-cCCTomics mediates efficient conditional knockout of chemoconnectome (CCT) genes. (**A**) Schematic of C-cCCTomics principle. Cas9 and three sgRNAs are driven by GAL4/UAS system. Three tandem sgRNAs are segregated by fly tRNA[Gly] and matured by RNase Z and RNase P. (**B–G**) Pan-neuronal knockout of Dh31 (**D**) and Pdf (**G**) by C-cCCTomics strategy. Representative fluorescence images presented expression of Dh31 (**B–D**) or

*Figure 2 continued on next page*

*Figure 2 continued*

anti-PDF (**E–G**). Pan-neuronal expression of Cas9 and sgRNA eliminated most (**D**) (n=6)or all (**G**) (n=7) fluorescent signal compared to control fly brains (**B–C, E–F**). Scale bar, 50 μm. (**H**) Activity profiles of pan-neuronal knockout of Pdfr and Pdfr-attpKO. Activity profiles were centered of the 12 hr darkness in all figures with evening activity on the left and morning activity on the right, which is different from general circadian literatures. Plotted in 30 min bins. (**I**) Statistical analysis of morning anticipation index (MAI), power, and period for pan-neuronal Pdf knockout and Pdfr-attpKO flies. Knocking out of Pdfr in neurons reduced both MAI, power, and period significantly. (**J, K**) Statistical analysis of nAChRα2 (**J**) and nAChRβ2 (**K**) pan-neuronal knockout flies' sleep phenotype. Sleep of these flies was not disrupted.

The online version of this article includes the following source data and figure supplement(s) for figure 2:

**Source data 1.** Data points for *Figure 2H–K*.

**Figure supplement 1.** Validation of primary C-cCCTomics.

**Figure supplement 1—source data 1.** Original files of the full raw unedited blots for *Figure 2—figure supplement 1B and F*.

**Figure supplement 1—source data 2.** Uncropped blots with the relevant bands labeled for *Figure 2—figure supplement 1B and F*.

efficiency (*Figure 3G, K, and L*), while Cas9.M6/sgRNA[ple] and Cas9.M9/sgRNA[ple] significantly improved efficiency to 87.53 ± 3.06% (n = 7) and 97.19 ± 2.15% (n = 8), respectively (*Figure 3I–L*). Fourteen additional CCT genes were subjected to pan-neuronal knockout, and the mRNA levels of the target genes were evaluated using real-time quantitative PCR with at least one primer overlapping the sgRNA targeting site (*Figure 3—figure supplement 1*). Cas9.M6 and Cas9.M9 demonstrated significantly higher gene disruption efficiency compared to the unmodified Cas9.HC, achieving average efficiencies of 87.51% ± 2.24% and 89.59% ± 2.39% for Cas9.M6 and Cas9.M9, respectively, in contrast to 70.72% ± 3.82% for Cas9.HC. (*Figure 3M*, *Figure 3—figure supplement 1*). To rule out the possibility of the observed variations in gene disruption efficiency being attributed to differential Cas9 expression levels, we quantified the Cas9 expression levels and noted that both Cas9.M6 and Cas9.M9 exhibited lower mRNA levels than Cas9.HC under the experiment condition (*Figure 3N*). Subsequently, genomic DNA of *Drosophila* head was extracted, and libraries encompassing target sites were constructed for high-throughput sequencing to verify disparities in genetic editing efficiency among these three Cas9 variants (*Figure 3O*). In almost all 19 sites tested, the mutation ratio consistently showed a trend toward Cas9.M6 and Cas9.M9 having a higher gene disruption efficiency than Cas9.HC (*Figure 3P*, *Figure 3—figure supplement 2*). The single-site mutation rates varied from 5.81% to 43.47% for Cas9.HC, 22.40% to 53.54% for Cas9.M6, and 19.90% to 63.57% for Cas9.M9 (*Figure 3P*, *Figure 3—figure supplement 2*). It should be noted that genomic DNA extracted from fly heads contained glial cells, which did not express Cas9/sgRNA, leading to a larger denominator and consequently reducing the observed mutation rates. Unmodified Cas9 displayed mutation rates comparable to those previously reported by *Schlichting et al., 2022*. The findings indicated that both Cas9.M6 and Cas9.M9 displayed elevated efficiency compared to Cas9.HC, with Cas9.M9 exhibiting the highest mutagenesis proficiency. These results suggest that the implementation of modified C-cCCTomics using Cas9.M6 and Cas9.M9 achieved an elevated level of efficiency. While unmodified C-cCCTomics was not efficient enough to knockout nAChRβ2 and nAChRα2 to phenocopy their mutants, we employed Cas9.M9 in cKO of these genes to verify its efficiency. Pan-neuronal knocking out of nAChRβ2 or nAChRa2 by Cas9.M9/sgRNA showed significant sleep decrease, which was similar to their mutants (*Figure 3Q, R*; *Dai et al., 2021*).

Taken together, our results support that we have created a high-efficiency toolkit for CCT gene manipulation in the nervous system, as well as more efficient Cas9 variants, Cas9.M6 and Cas9.M9, which can also be applied to genes other than those in the CCT.

## Forty-three CCT genes detected in clock neurons by genetic intersection

We analyzed the expression profile of CCT genes in circadian neurons with CCTomics driver lines in all clock neurons expressing Clk856 (*Gummadova et al., 2009*). With the Flp-out or split-LexA intersection strategy (*Figure 4A and B*), we found 43 out of 148 analyzed CCT genes expressed in circadian

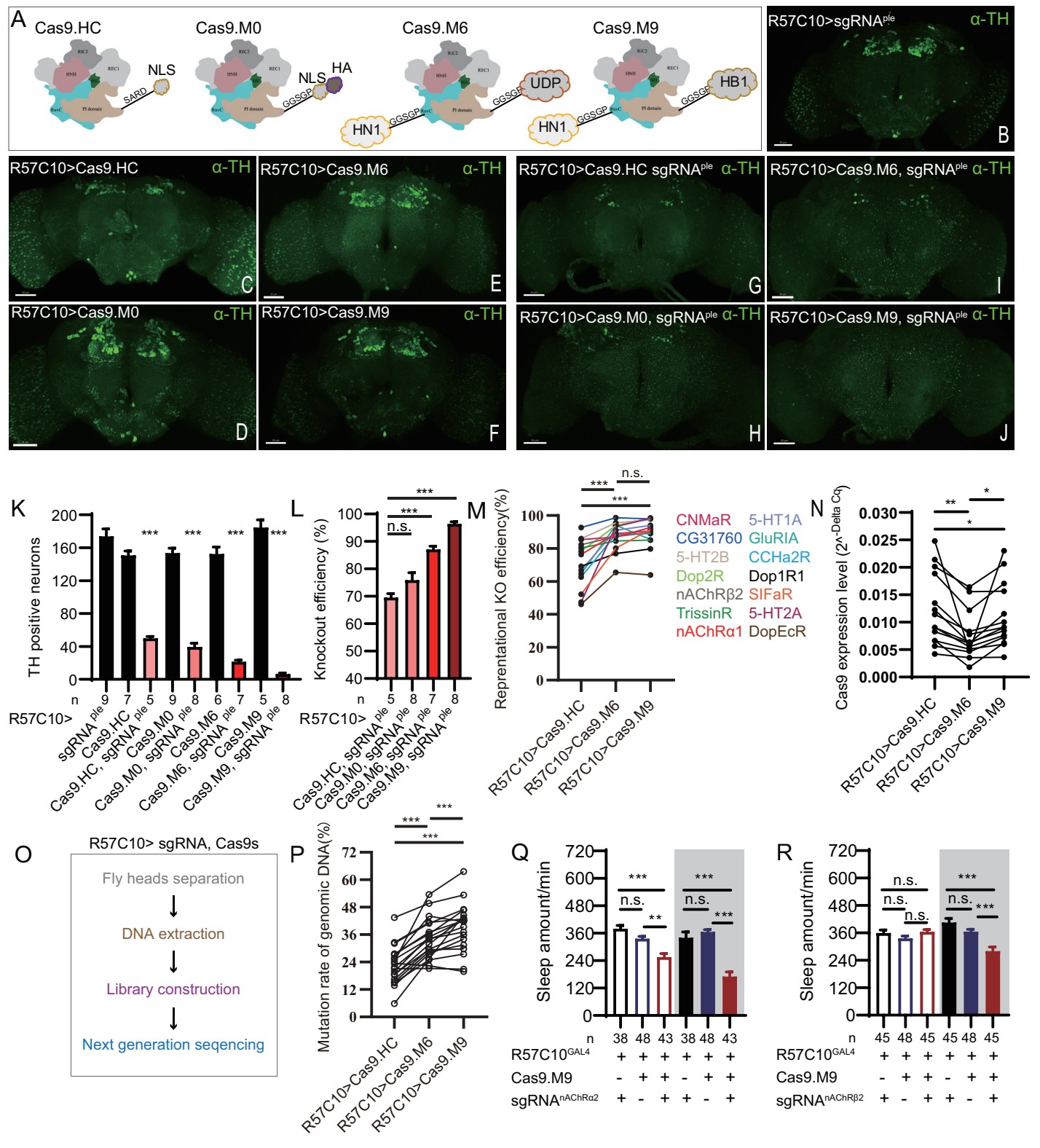

**Figure 3.** Efficiency evaluation of variations of chromatin-modulating peptides modified Cas9. (**A**) Schematics of chromatin-modulating peptides modified Cas9. (**B–J**) Efficiency evaluation of Cas9 variants. Fluorescence imaging of R57C10-Gal4>UAS-sgRNA^ple (**B**), R57C10-Gal4>UAS-Cas9 (**C–F**), and R57C10-Gal4>UAS-Cas9, UAS-sgRNA^ple (**G–J**) flies is shown. Brains were stained with anti-TH (green). Scale bar is 50 μm. (**K**) Anterior TH-positive neuron numbers of (**K–U**).The Kruskal–Wallis test followed by Dunn's post test was used, each Cas9 variant tested was compared to the two genotype controls (R57C10-GAL4/+; UAS-sgRNA^ple/+ and R57C10-GAL4/+; UAS-Cas9 variant/+ ) . ***p<0.001. (**L**) Statistical analysis of ple knockout efficiency

*Figure 3 continued on next page*

*Figure 3 continued*

related to (**K**). Modified Cas9.M6 and Cas9.M9 showed an improved efficiency compared to Cas9.HC. Student's *t*-test was used. (**M**) Statistical analysis of representational KO efficiency of Cas9 variants as related to *Figure 3—figure supplement 1*. Gene symbols on the right indicate tested genes. (**N**) Statistical analysis of Cas9 expression level. (**O, P**) Workflow of efficiency validation by next-generation sequencing (**O**) and statistical analysis of single-site mutation ratios induced by Cas9 variants (**P**). Paired *t*-test was used in (**M**), (**N**), and (**P**). (**Q, R**) Statistical analysis of sleep amount for nAChRα2 (**Q**) or nAChRβ2 (**R**) pan-neuronal knockout flies. Knockout of nAChRα2 and nAChRβ2 by modified Cas9.M9 significantly decreased flies' sleep amount.

The online version of this article includes the following source data and figure supplement(s) for figure 3:

**Source data 1.** Data points for *Figure 3K–O and Q–R*.

**Source data 2.** Original next-generation seq file for *Figure 3P*.

**Source data 3.** Original CRISPResso2 analysis report related to *Figure 3P* and *Figure 3—figure supplement 2*.

**Figure supplement 1.** Efficiency validation by real-time quantitative PCR.

**Figure supplement 1—source data 1.** Data points for *Figure 3—figure supplement 1*.

**Figure supplement 2.** Efficiency validation by high-throughput sequencing.

**Figure supplement 3.** Impact on viability of Cas9 variants expression by GMR57C10-GAL4.

**Figure supplement 3—source data 1.** Data points for *Figure 3—figure supplement 3*.

neurons (*Figure 4C*, *Figure 4—figure supplements 1–2*, and *Supplementary files 4 and 5* ). In all eight subsets of clock neurons, 23 CCT genes were expressed in DN1s, 20 in DN2s, 22 in DN3s, 28 in LNds, 14 in l-LNvs, 12 in s-LNvs, 5 in fifth s-LNv, and 3 in LPNs, with a total of 127 gene subsets.

To assess the accuracy of expression profiles using CCT drivers, we compared our dissection results with previous reports. Initially, we confirmed the expression of CCHa1 in two DN1s (*Fujiwara et al., 2018*), sNFP in four s-LNvs and two LNds (*Johard et al., 2009*), and Trissin in two LNds (*Ma et al., 2021*), aligning with previous findings. Additionally, we identified the expression of nAChRα1, nAChRα2, nAChRβ2, GABA-B-R2, CCHa1-R, and Dh31-R in all or subsets of LNvs, consistent with suggestions from studies using ligands or agonists in LNvs (*Duhart et al., 2020*; *Fujiwara et al., 2018*; *Lelito and Shafer, 2012*; *Shafer et al., 2008*; *Figure 4C* and *Supplementary file 4*).

Regarding previously reported Nplp1 in two DN1as (*Shafer et al., 2006*), we found approximately five DN1s positive for Nplp-KI-LexA, indicating a broader expression than previously reported. A similar pattern emerged in our analysis of Dh31-KI-LexA, where four DN1s, four s-LNvs, and two LNds were identified, contrasting with the two DN1s found in immunocytochemical analysis (*Goda et al., 2016*). Co-localization analysis of Dh31-KI-LexA and anti-PDF revealed labeling of all PDF-positive s-LNvs but not l-LNvs (*Figure 4—figure supplement 3A*), suggesting that the differences may arise from the broader labeling of 3' end knockin LexA drivers or the amplitude effect of the binary expression system. The low protein levels might go undetected in immunocytochemical analysis. This aligns with transcriptome analysis findings showing Nplp1 positive in DN1as, a cluster of CNMa-positive DN1ps, and a cluster of DN3s (*Ma et al., 2021*), which is more consistent with our dissection.

Despite the well-known expression of PDF in LNvs and PDFR in s-LNvs (*Shafer et al., 2008*), we did not observe stable positive signals for both in Flp-out intersection experiments, although both Pdf-KI-LexA and Pdfr-KI-LexA label LNvs as expected (*Figure 4—figure supplement 3B and C*). We also noted fewer positive neurons in certain clock neuron subsets compared to previous reports, such as NPF in three LNds and some LNvs (*Erion et al., 2016*; *He et al., 2013*; *Hermann et al., 2012*; *Johard et al., 2009*; *Lee et al., 2006*) and ChAT in four LNds and the 5th s-LNv (*Duhart et al., 2020*; *Johard et al., 2009*; *Supplementary file 4*). We attribute this limitation to the inefficiency of LexAop-FRT-myr::GFP driven by LexA, acknowledging that our intersection results may miss some positive signals.

## Conditional knockout of CCT genes in clock neurons

To investigate the function of CCT genes in circadian neurons with our cKO system, we knocked out all 67 (genes identified above and reported previously) CCT genes in Clk856-labeled clock neurons by C-cCCTomics.

In the pilot screen, we monitored fly activity by video recording (*Dai et al., 2019*) and analyzed rhythmic behavior under LD and DD conditions. We analyzed morning anticipation index (MAI) and evening anticipation index (EAI) under the LD condition (*Harrisingh et al., 2007*; *Im and Taghert, 2010*; *Seluzicki et al., 2014*; *Figure 5A*), power, period, and arrhythmic rate (AR) under the DD condition. Fly activities tended to rise rapidly after ZT22.5 at dawn and ZT10 at dusk. Thus, we added

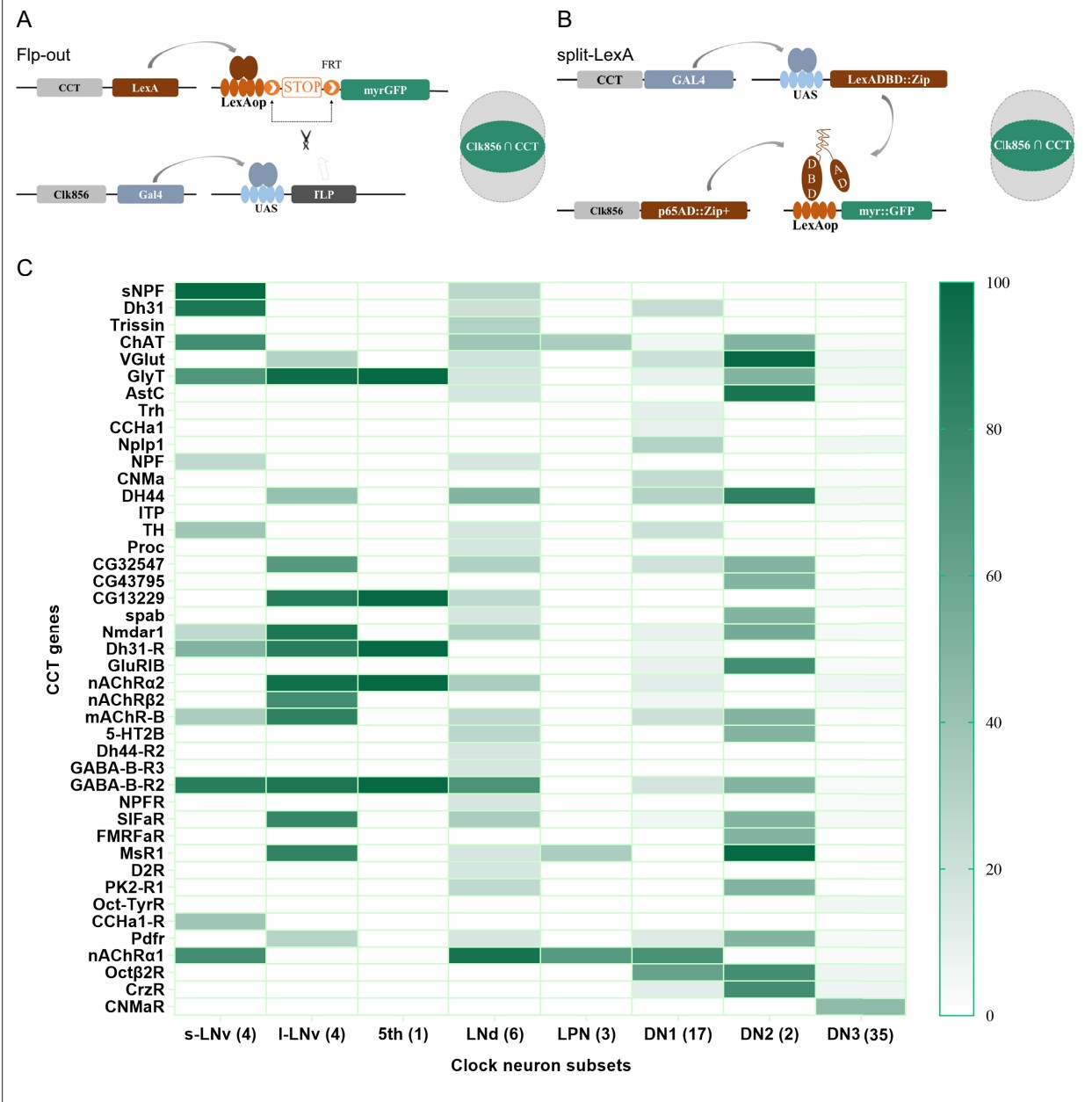

**Figure 4.** Genetic dissection of Clk856-labeled clock neurons. (**A, B**) Schematic of intersection strategies used in Clk856-labeled clock neurons dissection, Flp-out strategy (**A**) and split-LexA strategy (**B**). The exact strategy used for each gene is annotated in *Supplementary file 5*. (**C**) Expression profiles of CCT genes in clock neurons. Gradient color denotes proportion of neurons that were positive for the chemoconnectome (CCT) gene within each subset. The exact cell number for each subset is annotated in *Supplementary file 4*.

The online version of this article includes the following source data and figure supplement(s) for figure 4:

**Source data 1.** Exact percentage for *Figure 4C*.

**Figure supplement 1.** Co-expression of Clk856 with chemoconnectome (CCT) genes (01–23) intersectional expression patterns of CCT drivers with Clk856.

**Figure supplement 2.** Co-expression of Clk856 with chemoconnectome (CCT) genes (01–20) intersectional expression patterns of CCT drivers with Clk856.

**Figure supplement 3.** Co-localization analyses of Dh31, Pdfr, and Pdf -KI-LexA with LNvs.

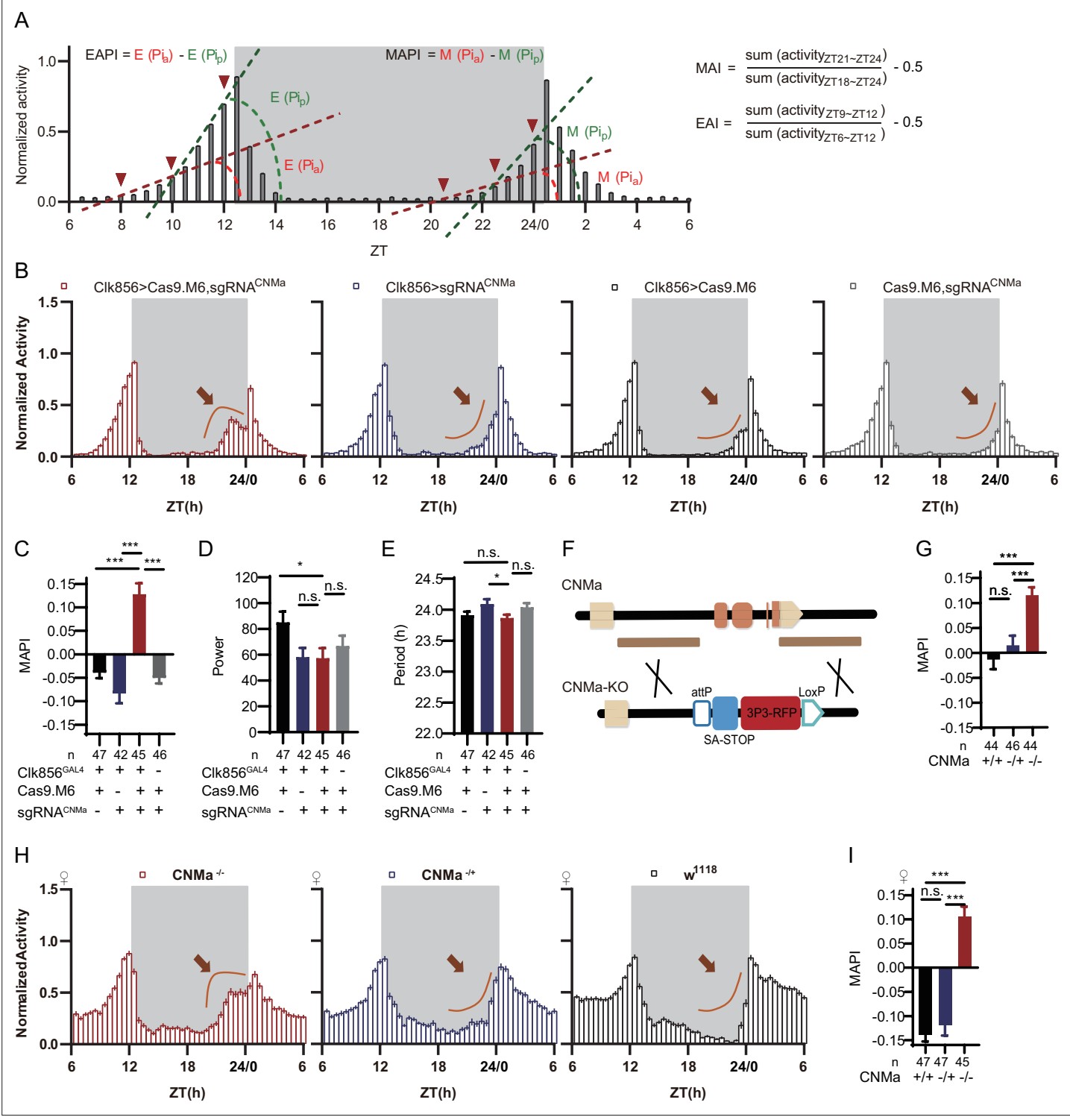

**Figure 5.** CNMa regulation of morning anticipation in clock neuron. (**A**) Schematic of morning anticipation index (MAI), evening anticipation index (EAI), morning anticipation pattern index (MAPI), and evening anticipation pattern index (EAPI) definition. (**B**) Activity plots of male flies with CNMa knockout in clock neurons (red) and controls (blue, black, and gray), plotted in 30 min bins. An advancement of morning activity peak was presented in CNMa clock neuron-specific mutants (brown arrowhead). (**C–E**) Statistical analyses of MAPI, power, and period of flies in (**B**). MAPI was significantly increased in clock neurons-specific CNMa-deficient flies (**C**) while power (**D**) and period (**E**) were not changed. (**F**) Schematic of CNMa[KO] generation. The entire encoding region of CNMa was replaced by an attP-SAstop-3P3-RFP-loxP cassette using CRISPR-Cas9 strategy. (**G**) Statistical analysis of MAPI of male CNMa[KO] flies (red) and controls (blue and black). MAPI significantly increased in male CNMa[KO] flies. (**H**) Activity plots of female CNMa[KO] flies (red) and

*Figure 5 continued on next page*

*Figure 5 continued*

controls (blue and black). (**I**) Statistical analysis of MAPI of female CNMa$^{KO}$ flies (red) and controls (blue and black). MAPI was significantly increased in female CNMa$^{KO}$ flies.

The online version of this article includes the following source data and figure supplement(s) for figure 5:

**Source data 1.** Data points for *Figure 5B–E and G–I*.

**Figure supplement 1.** Disruption of chemoconnectome (CCT) genes due to leaky expression of Cas9.M9.

**Figure supplement 1—source data 1.** Data points for *Figure 5—figure supplement 1A–D*.

**Figure supplement 2.** Morning activity advanced by loss of CNMa.

**Figure supplement 2—source data 1.** Data points for *Figure 5—figure supplement 2A–G*.

two more parameters to describe the anticipatory activity patterns of LD condition. Morning anticipation pattern index (MAPI) was defined as the difference between $Pi_a$[arctan(ZT20.5~ZT22.5 activity increasing slope)] and $Pi_p$[arctan(ZT22.5~ZT24 activity increasing slope)], M($Pi_a$-$Pi_p$). Evening anticipation pattern index (EAPI) was defined similar to MAPI (*Figure 5A*, see 'Materials and methods'). $Pi_a$ and $Pi_p$ were positive, while MAPI and EAPI were negative, for wild type (wt) flies as their activities gradually increases at dawn or dusk at increasing rates.

Knocking out Pdf or Pdfr in clock neurons phenocopied their mutants with lower MAI, advanced evening activity, low power, high arrhythmic rate, and shorter period (*Hyun et al., 2005*; *Lear et al., 2005*; *Renn et al., 1999*: *Supplementary file 6* and *Figure 5—figure supplement 1A–D*). The MAI-decreasing phenotype of Dh31 knockout was also reproduced in this pilot screen (*Goda et al., 2019*; *Supplementary file 6*). All the above results verified the effectiveness of C-cCCTomics. Unexpectedly, additional experimental replications with full controls using Cas9.M9 revealed that leaky expression of Cas9.M9 and sgRNA might have caused disruption of Dh31, Dh44, Pdf, and Pdfr (*Figure 5—figure supplement 1A–D*), which was not suitable for neuronal-specific mutagenesis of some genes. Therefore, in the following work we primarily focused on Cas9.M6 instead.

Analysis of the newly defined parameters MAPI and EAPI showed that control flies (Clk856-GLA4>UAS-Cas9.M9) had negative EAPI but slightly positive MAPI. The positive MAPI of control flies in this screen might be caused by Cas9.M9 toxicity. Only the Pdf and Pdfr clock neuron knockout flies showed positive EAPIs, indicating an advanced evening activity (*Supplementary file 6* and *Figure 5—figure supplement 1A and C*). nAChRα1, MsR1, mAChR-B, and CNMa cKO flies had the highest MAPI values (*Supplementary file 6*). We further confirmed their phenotypes using Cas9.M6, which revealed that CNMa plays a role in regulating morning anticipatory activity (*Figure 5—figure supplement 2A*).

## Regulation of morning anticipation by CNMa-positive DN1p neurons

Conditionally knocking out CNMa in clock neurons advanced morning activity (*Figure 5B*, *Figure 5—figure supplement 2B*) and increased MAPI (*Figure 5C*, *Figure 5—figure supplement 2A and C*), leaving the power and period intact in male flies (*Figure 5D and E*). The same advanced morning activity phenotype was also observed in female flies (*Figure 5—figure supplement 2D–G*). To further validate this phenotype, we generated a CNMa knockout (CNMa$^{KO}$) line by replacing its whole coding region with an attP-splicing adaptor element (*Figure 5F*). Both male and female CNMa$^{KO}$ flies exhibited the same phenotypes as seen in the CNMa cKO (*Figure 5G–I*).

Previous studies have found CNMa expression in DN1 neurons (*Abruzzi et al., 2017*; *Jin et al., 2021*; *Ma et al., 2021*). Our intersection showed four DN1p and one DN3 CNMa-positive neurons in Clk856-labeled neurons (*Figure 4—figure supplement 2*, #16, *Figure 4C*). Analysis with an endogenous CNMa-KI-GAL4 knockin driver showed that six pairs of CNMa neurons located in the DN1p region and three pairs located in the subesophageal ganglion (SOG) had the brightest GFP signals (*Figure 6A*). The anatomical features of CNMa neurons were further confirmed using stingerRed and more neurons were found in regions, the anterior ventrolateral protocerebrum (AVLP), and the antennal mechanosensory and motor center (AMMC) (*Figure 6—figure supplement 1A*). Dendrites of CNMa neurons were concentrated in DN1p and SOG, with their axons distributed around DN1p region, lateral horn (LH), and prow region (PRW) (*Figure 6B and C*). Using the trans-tango strategy (*Talay et al., 2017*), we also found that downstream of CNMa neurons were about 15 pairs of neurons in the

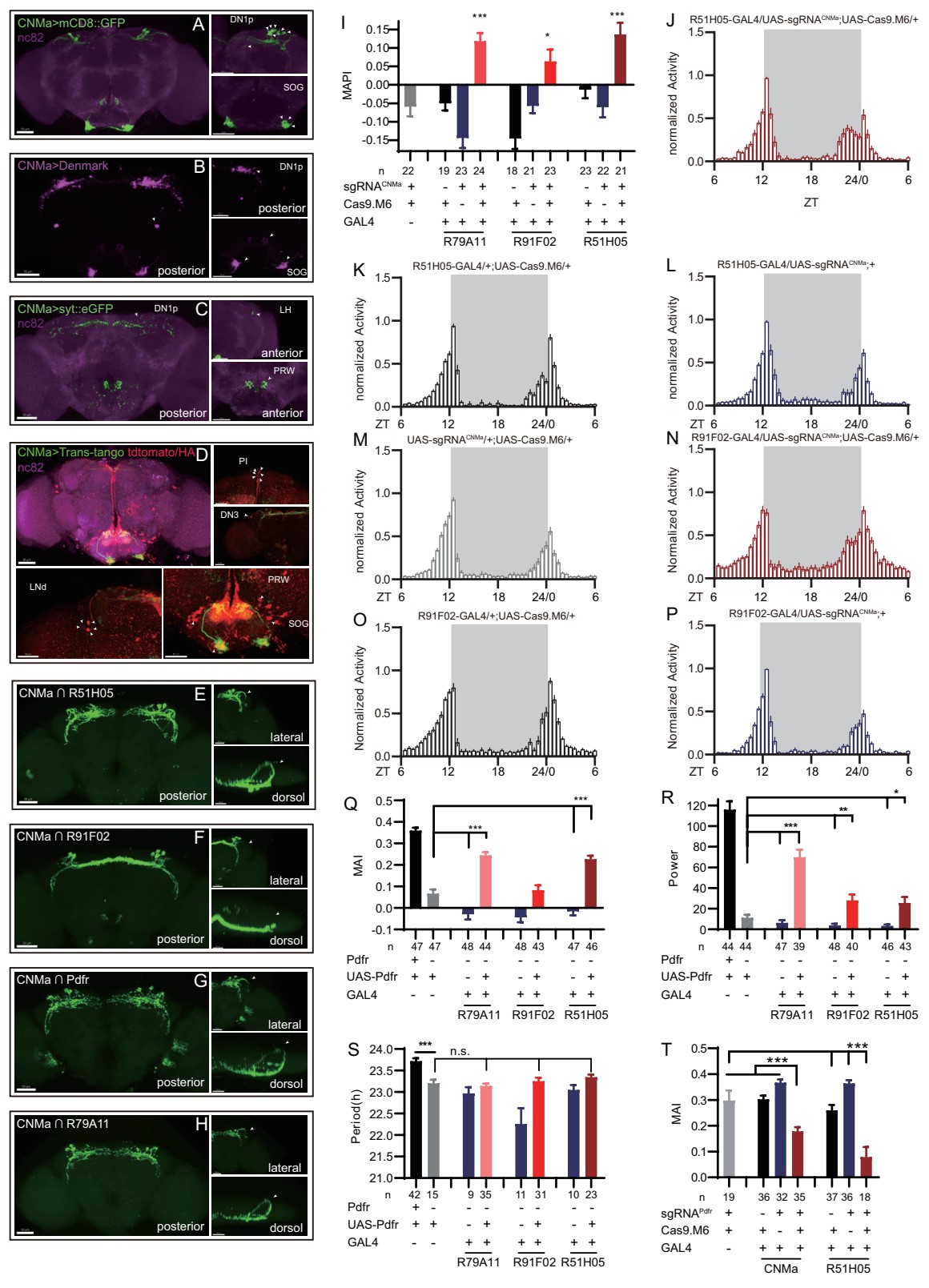

**Figure 6.** Expression, projection, and trans-projection feature of CNMa neurons and its functional subset. (**A–C**) Expression and projection patterns of CNMa-KI-Gal4 in the brain. Membrane, dendrites, and axon projections are labeled by mCD8::GFP (**A**), Denmark (**B**), and syt::eGFP (**C**), respectively. (**D**) Downstream neurons labeled through trans-tango driven by CNMa-KI-GAL4. Arrowheads indicate candidate downstream neurons: 6 neurons in PI, 1 pair in DN3, 5 pairs in LNd, and about 15 pairs in subesophageal ganglion (SOG). (**E–H**) Intersection of DN1p CNMa neurons with DN1p-labeled drivers.

*Figure 6 continued on next page*

*Figure 6 continued*

GMR51H05-GAL4 (**E**), GMR91F02-GAL4 (**F**), Pdfr-KI-GAL4 (**G**), and GMR79A11-GAL4 (**H**) were intersected with CNMa-p65AD, UAS-LexADBD, LexAop-myr::GFP. Two type I (**E, G, H**) neurons projected to anterior region and four type II (**F**) neurons had fewer projections to anterior region. Scale bar, 50 µm. (**I**) Morning anticipation pattern index (MAPI) was significantly increased in all three DN1p drivers-mediated CNMa knockout. Each experimental group(red) was compared to their three genotype controls. (**J, K**) Activity plots of CNMa knockout in R51H05-GAL4 (**J**) and R91F02-GAL4 (**N**) neurons. R51H05-GAL4-mediated CNMa knockout flies showed an advanced morning activity peak (**J**), while R91F02-Gal4-mediated CNMa knockout flies did not (**N**). (**Q–S**) Statistical analyses of morning anticipation index (MAI), power, and period. Each experimental group (pink, light red and dark red)  was compared to their genotype controls (grey and blue).There was no significant difference between Pdfr-attpKO;UAS-Pdfr/+; R91F02-GAL4/+(light red) and its genotype control Pdfr-attpKO;UAS-Pdfr/+ (grey). Pdfr reintroduction in R79A11 and R51H05 neurons could partially rescue the MAI-decreased phenotype of Pdfr knockout flies. (**T**) Statistical analyses of MAI of Pdfr knocking out in CNMa-GAL4 and R51H05-GAL4-labeled neurons.

The online version of this article includes the following source data and figure supplement(s) for figure 6:

**Source data 1.** Data points for *Figure 6I–T*.

**Figure supplement 1.** Expression of CNMa and CNMaR.

**Figure supplement 2.** Activity plots related to *Figure 6*.

**Figure supplement 2—source data 1.** Data points for *Figure 6—figure supplement 2A and B*.

SOG, 5 pairs of LNd neurons, 1 pair of DN3 neurons, and 6 intercerebralis (PI) neurons (*Figure 6D*, arrowhead).

Because we had found that knocking out CNMa in Clk856-GAL4-labeled neurons produced advanced morning activity, and that CNMa intersected with Clk856-Gal4-labeled neurons in four pairs of DN1ps and one pair of DN3 neurons (*Figure 4—figure supplement 2*, #16), we focused on these neurons and performed more intersections. Taking advantage of a series of clock neuron subset-labeled drivers (*Sekiguchi et al., 2020*), we intersected CNMa-p65AD with four DN1 labeling drivers: GMR51H05-GAL4, GMR91F02-GAL4, Pdfr-KI-GAL4, and GMR79A11-GAL4 (*Figure 6E–H*). We found two arborization patterns: type I with two neurons whose branches projecting to the anterior region, as observed in CNMa∩GMR51H05, CNMa∩Pdfr, and CNMa∩GMR79A11 (*Figure 6E, G, and H*), and type II with four neurons branching on the posterior side with few projections to the anterior region, as observed in CNMa∩GMR91F02 (*Figure 6F*). These two types of DN1ps' subsets have been previously reported and profoundly discussed (*Lamaze et al., 2018*; *Reinhard et al., 2022*).

CNMa knockout in type I or type II neurons (GMR51H05-GAL4, GMR91F02-GAL4, and GMR79A11-GAL4) all reproduced the MAPI-increased phenotype of clk856-specific CNMa knockout (*Figure 6I*). However, type II neurons-specific CNMa knockout (CNMa ∩ GMR91F02) showed weaker advanced morning activity without advanced morning peak (*Figure 6N*), while type I neurons-specific CNMa knockout did (*Figure 6J*), indicating a possibility that these two type I CNMa neurons are the main functional subset regulating the morning anticipation activity of fruit fly.

Pdf or Pdfr mutants exhibit weak or no morning anticipation, which is related to the phenotype of CNMa knockout flies. We also identified two Pdfr and CNMa double-positive DN1ps, which have a type I projection pattern (*Figure 6G*). Reintroduction of Pdfr in Pdfr knockout background revealed that GMR51H05 and GMR79A11 Gal4 drivers, which covered the main functional CNMa-positive subset, could partially rescue the morning anticipation and power phenotype of Pdfr knockout flies to a considerably larger extent than the GMR91F02 driver (*Figure 6Q–S*, *Figure 6—figure supplement 2A*, and *Supplementary file 7*). Moreover, knocking out Pdfr by GMR51H05 and CNMa GAL4, which cover type I CNMa neurons, decreased morning anticipation of flies (*Figure 6T*, *Figure 6—figure supplement 2B*). However, the decrease in morning anticipation observed in the Pdfr knockout by CNMa-GAL4 was not as pronounced as with GMR51H05-GAL4. Because the presumptive main subset of functional CNMa is also PDFR-positive, there is a possibility that CNMa secretion is regulated by PDF/PDFR signal.

## Role of neuronal CNMaR in morning anticipation

There is only one CNMa receptor reported in the fly genome (*Jung et al., 2014*). We generated a CNMaR[KO-p65AD] line by CRISPR/Cas9 (*Figure 7A*), and this knockout showed advanced morning activity (*Figure 7B and D*) and increased MAPI (*Figure 7C and E*) in both sexes. CNMaR[KI-Gal4]/UAS-mCD8::GFP and CNMaR[KI-Gal4]/UAS-stinger::Red showed expression of CNMaR across the whole brain (*Figure 7F*, *Figure 6—figure supplement 1B*), especially in DN1p, DN3, the PI, and the SOG. The dendrite arborization and synaptic projections of CNMaR neurons also covered broad regions (*Figure 7G and*

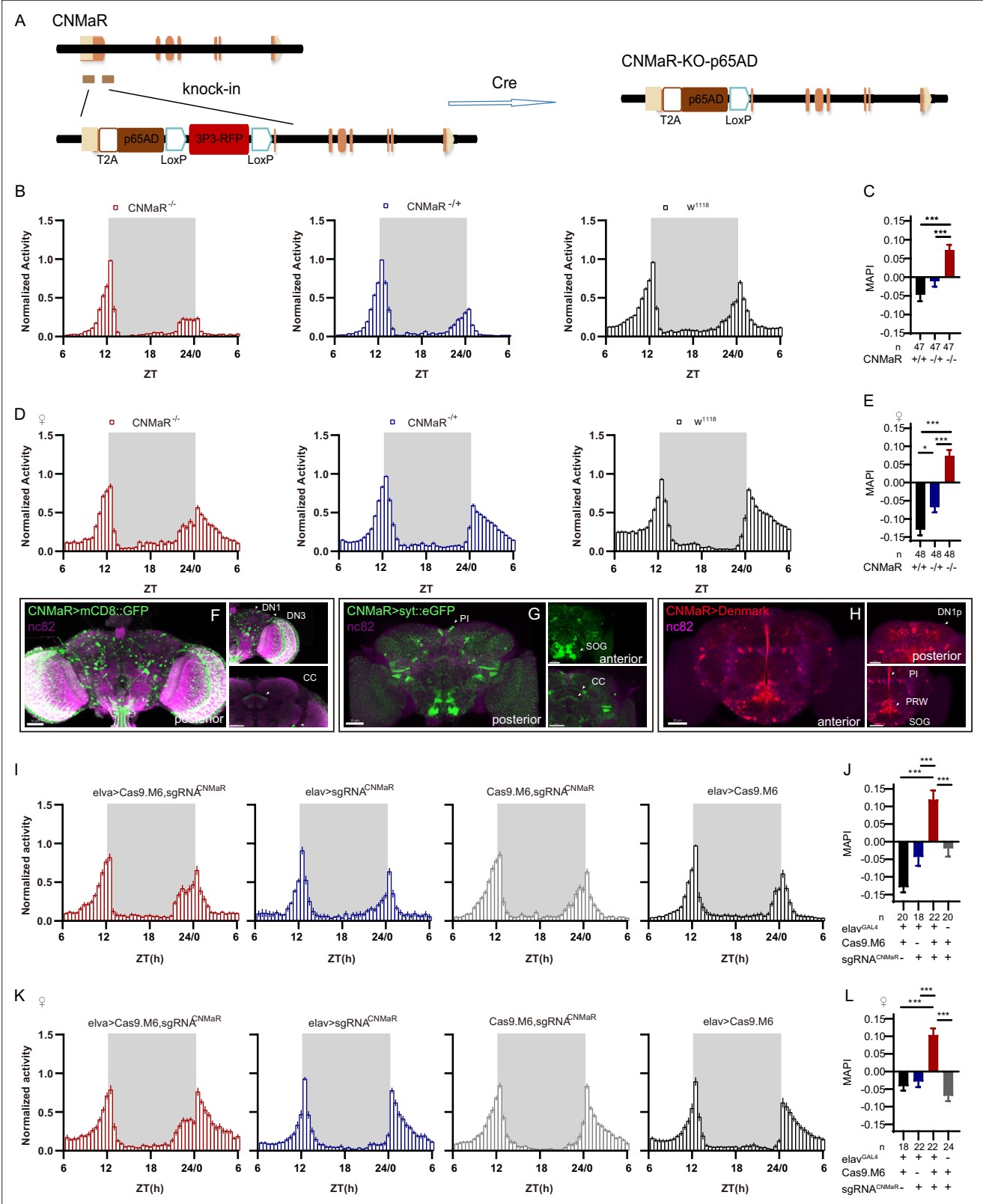

**Figure 7.** CNMaR regulation of morning anticipation. (**A**) Schematic of CNMaR^KO-p65AD generation. Most of the first exon in CNMaR was replaced by a T2A-p65AD-loxP-3P3-RFP-loxP cassette using CRISPR-Cas9 strategy and the T2A-p65AD was inserted in the reading frame of the remaining CNMaR codon. 3P3-RFP was removed latterly by Cre mediated recombination. (**B–E**) Activity plot (**B, D**) and statistical analysis (**C, E**) of male (**B–C**) or female (**D–E**) CNMaR^KO-p65AD flies (red) and genotypical controls (blue and black). Morning anticipation pattern index (MAPI) was significantly increased in both

*Figure 7 continued on next page*

*Figure 7 continued*

male and female CNMaR^KO-p65AD flies. In this and other figures, '♀' denotes female flies. (**F–H**) Expression and projection patterns of CNMaR-KI-Gal4 in the brain. Scale bars, 50 μm. (**I–L**) Activity plots (**I, K**) and statistical analyses (**J, L**) of CNMaR pan-neuronal knockout flies. Neuronal knockout of CNMaR increased MAPI (**J, L**).

The online version of this article includes the following source data for figure 7:

**Source data 1.** Data points for *Figure 7B–E and I–L*.

*H*), at the PI, the SOG, the posterior ventrolateral protocerebrum (PVLP), and the central complex (CC). Further cKO of CNMaR in neurons by C-cCCTomics phenocopied CNMaR^KO-p65AD phenotype (*Figure 7I–L*). These results indicate that CNMaR is similar to CNMa in regulating morning anticipation.

## Discussion
### Conditional CCTomics strategies and toolkit
We have generated conditional gene manipulation systems based on Flp-out/GFPi or CRISPR/Cas9. cCCT-based gene deletion after heat-shock or mifepristone (RU486) eliminated most GFP signals, and pan-neuronal constitutive expression of shRNA^GFP or flippase disrupted seven out of eight tested genes completely while targeting of SIFa^EGFP.FRT achieved 96% ± 3% efficiency. Although the recombination of genetic elements is relatively cumbersome when using cCCTomics, it is worthwhile applying this method to specific genes given its high level of efficiency. While two UAS-sgRNA libraries have been established, one primarily targeting kinases (*Port et al., 2020*) and the other targeting GPCRs (*Schlichting et al., 2022*), both libraries only cover a portion of CCT genes, and are thus insufficient for manipulating all CCT genes. The development of C-cCCTomics, however, makes CCT gene manipulation as simple as RNA interference. Furthermore, the use of modified Cas9.M6 or Cas9.M9 highly enhances the efficiency of gene disruption in the nervous system, allowing for efficient manipulation of all CCT genes in a cell-specific manner.

The toxicity of CRISPR/Cas9 depends on the Cas9 protein (*Port et al., 2014*). When expressed pan-neuronally in nSyb-GAL4 (R57C10-GAL4, attP40), Cas9.M9 slightly reduced viability, while the expression of other Cas9 variants had no significant effect on viability (*Figure 3—figure supplement 3*). Although Cas9.M9 showed leaky expression efficiency, this was not a problem with Cas9.M6, which successfully disrupted Dh31, Dh44, Pdf, and Pdfr (*Figure 5—figure supplement 1*). A more restricted expression of Cas9.M9 with lower toxicity is necessary for better somatic gene manipulation in the future.

### CCT of clock neurons
Intersecting Clk856-Gal4 or Clk856-p6AD with CCTomics, we identified 43 CCT genes in Clk856-labeled clock neurons. Clock neurons appear highly heterogeneous both in our intersection dissection and in a previous transcriptomic analysis (*Abruzzi et al., 2017*; *Ma et al., 2021*). Comparing these two CCT gene expression profiles in clock neurons, 41 out of 127 gene subsets are identical. The accuracy of our genetic intersection is limited by two possibilities: (1) KI-LexA may not fully represent the expression pattern of the corresponding gene, and (2) the efficiency of STOP cassette removal in the Flp-out strategy is limited or the efficiency of LexA>LexAop::myrGFP. Moreover, the leakage of LexAop-GFP may result in unreliable labeling in split-LexA strategy. Both genetic drivers and transcriptomic analysis contribute to our knowledge of the expression profile of neurons. The physiological significance of each gene in particular neurons should be further investigated by genetic manipulation.

### Regulation of rhythmic behavior by CCT genes
Multiple attractive genes have been identified in our functional screen of CCT genes in clock neurons: for example, knocking out of VGlut weakens morning anticipation (*Supplementary file 8*). In further screening of brain regions, we have narrowed down the morning anticipation regulation role of VGlut in R18H11-GAL4-labeled neurons (*Supplementary file 9*). VGlut in these neurons has also been reported to regulate sleep in *Drosophila* (*Guo et al., 2016*). Its downstream neurons may be the PI neurons or LNvs (*Barber et al., 2021*; *Guo et al., 2016*).

Moreover, the deficiency of the neuropeptide CNMa results in advanced morning activity. We have validated that two Pdfr and CNMa double-positive DN1p neurons may play a major role in regulating this process through intersectional manipulation of CNMa. Knockout and reintroduction of Pdfr in these neurons have verified that Pdfr partially functions in DN1p CNMa neurons, and PDF increases cAMP level in Pdfr-positive neurons (*Shafer et al., 2008*), suggesting a possibility of the regulation of CNMa signaling by PDF signaling. Furthermore, given that the morning anticipation vanishing phenotype of Pdf or Pdfr mutant indicates a promoting role of PDF-PDFR signal, while the enhanced morning anticipation phenotype of CNMa mutant suggests an inhibiting role of CNMa signal, we consider the two signals to be antagonistic. However, knocking out CNMaR in Clk856-labeled clock neurons showed no significant phenotype (*Supplementary file 6*), whereas the mutant and pan-neuronal knockout flies had similar phenotypes to CNMa knockout flies, suggesting its role in the circadian output neurons. Previous studies have indicated that CNMa integrate thermosensory inputs to promote wakefulness, and CNMaR is thought to function in Dh44-positive PI neurons (*Jin et al., 2021*), a subset of circadian output neurons. To gain a deeper understanding of the downstream effects of DN1p CNMa-positive neurons, further analysis focusing on specific brain regions is necessary.

We have also reproduced phenotypes of Pdf, Pdfr, and Dh31 mutant flies with C-cCCTomics as previous studies. Surprisingly, only 5 genes are functional among all 67 CCT genes in this prior screen. This may be caused by limitations of the simple behavioral paradigm, single-gene manipulation, and single GAL4 driver. For example, switching of light condition from L:D = 12 hr:12 hr to L:D = 6 hr:18 hr, AstC/AstC-R2 would suppress flies' evening activity intensity to adapt to the environmental change (*Díaz et al., 2019*), and only double knockout of AChRs and mGluRs in PI neurons can possibly result in alteration in behavioral rhythms (*Barber et al., 2021*). Further diversified functional analysis of CCT genes in clock neurons is required for clock circuit dissection.

# Materials and methods

**Key resources table**

| Reagent type (species) or resource | Designation | Source or reference | Identifiers | Additional information |
|---|---|---|---|---|
| Antibody | Anti-Bruchpilot antibody (mouse monoclonal) | Developmental Studies Hybridoma Bank | RRID:AB_2314866 | 1:40 |
| Antibody | Anti-TH (rabbit polyclonal) | Novus Biologicals | NOVUS NB300-109; RRID:AB_10077691 | 1:1000 |
| Antibody | Anti-mouse IgG-Alexa633 (goat polyclonal) | Invitrogen | Cat#A-21050; RRID:AB_2535718 | 1:1000 |
| Antibody | Anti-PDF (mouse monoclonal) | Developmental Studies Hybridoma Bank | PDF C7; RRID:AB_760350, AB_2315084 | 1:200 |
| Antibody | Anti-LK (rabbit polyclonal) | Rao Lab | Anti-LK | 1:1000 |
| Antibody | Anti-DSK (rabbit polyclonal) | *Wu et al., 2020* | Anti-DSK | 1:1000 |
| Antibody | Anti-rabbit IgG- Alexa488 (goat polyclonal) | Invitrogen | Cat# A-11008; RRID:AB_143165 | 1:1000 |
| Antibody | Anti-rabbit IgG- Alexa633 (goat polyclonal) | Invitrogen | Cat# A-21070; RRID:AB_2535731 | 1:1000 |
| Genetic reagent (*Drosophila melanogaster*) | CCT attP KO lines | *Deng et al., 2019* | N/A | |
| Genetic reagent (*D. melanogaster*) | CCT KI GAL4/LexA lines | *Deng et al., 2019* | N/A | |
| Genetic reagent (*D. melanogaster*) | Clk856-GAL4 | Bloomington | #93198 | |
| Genetic reagent (*D. melanogaster*) | w*, P{nos-phiC31\int.NLS}X;;P{CaryP}attP2 | Jenelia Research Campus | N/A | |

*Continued on next page*

*Continued*

| Reagent type (species) or resource | Designation | Source or reference | Identifiers | Additional information |
|---|---|---|---|---|
| Genetic reagent (*D. melanogaster*) | w[1118];P{GMR57C10-GAL4}attP40 | Luo Lab, Peking University | N/A | |
| Genetic reagent (*D. melanogaster*) | y[1] M{vas-int.Dm}ZH-2A w[*]; PBac{y+-attP-9A}VK00005 | Bloomington | #24862 | |
| Genetic reagent (*D. melanogaster*) | y[1] w[*] P{nos-phiC31\int.NLS}X; P{CaryP}attP40 | Bloomington | #79604 | |
| Genetic reagent (*D. melanogaster*) | y1 w*; P{UAS-mCD8::GFP.L}LL5, P{UAS-mCD8::GFP.L}2 | Bloomington | #5137 | |
| Genetic reagent (*D. melanogaster*) | w[1118];P{GMR57C10-GAL4}attP2 | Bloomington | #39171 | |
| Genetic reagent (*D. melanogaster*) | 20xUAS-IVS-FLP1;in attp2 | Jenelia Research Campus | 1116428 PJFRC152 | |
| Genetic reagent (*D. melanogaster*) | 20xUAS-IVS-FLP1;;PEST in attp2 | Jenelia Research Campus | 1116430 PJFRC150 | |
| Genetic reagent (*D. melanogaster*) | y[1] sc[*] v[1]; P{y[+t7.7] v[+t1.8]=VALIUM20-EGFP.shRNA.1}attP40 | Bloomington | #41555 | |
| Genetic reagent (*D. melanogaster*) | P{13XLexAop2(FRT.stop)myr::GFP}attP2 | Rubin Lab | #1116847 | |
| Genetic reagent (*D. melanogaster*) | y1 w*; P{elav-Switch.O}GSG301 | Bloomington | #43642 | |
| Genetic reagent (*D. melanogaster*) | w*; P{trans-Tango}attP40 | Bloomington | #77123 | |
| Genetic reagent (*D. melanogaster*) | w[1118]; P{w[+mC]=UAS-RedStinger}4/CyO | Bloomington | #8546 | |
| Genetic reagent (*D. melanogaster*) | w[1118]; P{w[+mC]=UAS-Denmark}2 | Bloomington | #33062 | |
| Genetic reagent (*D. melanogaster*) | w[*]; P{w[+mC]=UAS-syt.eGFP}3 | Bloomington | #6926 | |
| Genetic reagent (*D. melanogaster*) | UAS-Cas9.HC(VK00005) | This paper | | Rao Lab |
| Genetic reagent (*D. melanogaster*) | UAS-Cas9.M0(attP40 or attP2) | This paper | | Rao Lab |
| Genetic reagent (*D. melanogaster*) | UAS-Cas9.M6(attP40 or attP2) | This paper | | Rao Lab |
| Genetic reagent (*D. melanogaster*) | UAS-Cas9.M9(attP40 or attP2) | This paper | | Rao Lab |
| Recombinant DNA reagent | pACU2 | Jan Lab, University of California, San Francisco | N/A | |
| Recombinant DNA reagent | pEC14 | *Deng et al., 2019* | N/A | |
| Recombinant DNA reagent | pBSK-attB-loxP-myc-T2A-Gal4-GMR-miniwhite | *Deng et al., 2019* | N/A | |
| Recombinant DNA reagent | pAAV-Efla-DIO-mScarlet | Addgene | #130999 | |
| Software, algorithm | MATLAB | MathWorks, Natick, MA | https://www.mathworks.com/products/matlab.html | |

*Continued on next page*

Continued

| Reagent type (species) or resource | Designation | Source or reference | Identifiers | Additional information |
|---|---|---|---|---|
| Software, algorithm | Prism 8 | GraphPad | https://www.graphpad.com/ | |
| Software, algorithm | Imaris | Bitplane | http://www.bitplane.com/imaris/imaris | |

## Fly lines and rearing conditions

Flies were reared on standard corn meal at 25°C, 60% humidity, 12 hr light:12 hr dark (LD) cycle. For flies used in behavior assays, they were backcrossed into our isogenized Canton S background for 5–7 generations. For heat-induced assays, flies were reared at 20°C. All CCT attP KO lines and CCT KI driver lines were previous generated at our lab (*Deng et al., 2019*). Clk856-GAL4 and GMR57C10-GAL4 driver lines were gifts from Donggen Luo Lab (Peking University). 13XLexAop2 (FRT.stop) myr::GFP was a gift from Rubin Lab.

## C-cCCTomics sgRNAs design

All sgRNAs target at or before functional coding regions (e.g., GPCR transmembrane domain, synthetase substrate binding domain) of each CCT genes. For each gene, about 20 sgRNAs with specific score ≥12 were firstly designed at the CRISPRgold website (*Chu et al., 2016*; *Graf et al., 2019*), then their specificity and efficacy were further valued in Optimal CRISPR target finder (*Gratz et al., 2014*), E-CRISPR (*Heigwer et al., 2014*), and CCTop (*Stemmer et al., 2015*) system. The first three highest efficacy sgRNAs with no predicted off-target effect were selected. All selected sgRNAs are listed in *Supplementary file 3*.

## Molecular biology

All cCCTomics knockin (KI) lines and C-cCCTomics transgenic flies were generated through phiC31-mediated attB/attP recombination, and the miniwhite gene was used as selection marker.

For cCCTomics KI lines, backbone pBSK-attB-FRT-*HpaI*-T2A-EGFP-FRT was modified from pBSK-attB-loxP-myc-T2A-Gal4-GMR-miniwhite (*Deng et al., 2019*). Myc-T2A-GAL4 cassette was removed by PCR amplification while first FRT cassette was introduced. Second FRT cassette was inserted by T4 ligation between *SpeI* and *BamHI*. T2A-EGFP was cloned from pEC14 and was inserted into the backbone between two FRT cassettes. All gene spans, except for stop codon, deleted in CCT attP KO lines were cloned into pBSK-attB-FRT-*HpaI*-T2A-EGFP-FRT at *HpaI* site (*Supplementary file 10*).

For C-cCCTomics UAS-sgRNA lines, backbone pMsgNull was modified from pACU2 (*Han et al., 2011*). Synthetic partial fly tRNA$^{Gly}$ sequence was inserted between *EcoRI* and *KpnI*. An irrelevant 1749 bp cassette amplified from pAAV-Efla-DIO-mScarlet (Addgene #130999) was inserted between *EagI* and *KpnI*. All sgRNA spacers were synthesized at primers, and 'E+F' sgRNA scaffold and rice tRNA$^{Gly}$ was amplified from a synthetic backbone PM04. Finally, gRNA-tRNA$^{Gly}$ cassettes were cloned into pMsgNull between *EagI* and *KpnI* by Gibson Assembly (*Supplementary file 10*).

All UAS-Cas9 variants generated in this research were cloned into vector pACU2 (between EcoRI and Acc65I) and all Cas9 sequences were amplified from hCas9 (Addgene #41815). Human codon-optimized Cas9 was cloned into pACU2 to generate UAS-Cas9.HC. UAS-Cas9.M0 was modified from UAS-Cas9.HC by introducing an HA tag after NSL and replacing the SARD linker with the GGSGP linker (*Zhao et al., 2016*). UAS-Cas9.M6 and UAS-Cas9.M9 were designed as HMGN1-Cas9-UPD and HMGN1-Cas9-HB1, respectively. All these chromatin-modulating peptides were linked with Cas9 by GGSGP linker (*Supplementary file 10*).

CNMaR$^{KO-p65AD}$ was generated by replacing the coding region of the first exon with T2A-p65AD by CRISPR/Cas9, and the T2A-p65AD was linked in frame after first 10 amino acids. Spacers of gRNAs used to break the targeted CNMaR region were 5'-GCAGATTTCAGTTCATCTTT-3', 5'-GGCTTGGCAATGACTATATA-3'.

## Gene expression quantitation and high-throughput sequencing

Female flies were gathered 6–8 d post-eclosion for gene expression quantification and high-throughput sequencing. Fly heads were isolated by chilling them on liquid nitrogen and subsequent shaking.

mRNA extraction was performed using Trizol according to a previously established protocol (**Green and Sambrook, 2020**). Genomic DNA was removed, and cDNA was synthesized using a commercial kit (TIANGEN#DP419). For real-time quantitative PCR, at least one PCR primer was designed to overlap with the sgRNA target site.

Genomic DNA from fly heads was extracted using a standard alkali lysis protocol (**Huang et al., 2009**). Genomic regions approximately 130–230 bp in length, centered around the sgRNA target site, were amplified by PCR employing Q5 polymerase (NEB#M0494). Subsequently, libraries were prepared using the BTseq kit (Beijing Tsingke Biotech Co., Ltd). These libraries were pooled and subjected to sequencing on the MiSeq platform (Illumina). Analysis of the libraries was conducted using Crispresso2 (**Clement et al., 2019**).

## Generation of KI and transgenic lines

Generation of cCCTomics KI, CNMa$^{KI-p65AD}$, and CNMaR$^{KO-p65AD}$ lines are the same as the generation of CCTomics KI driver lines as previously described (**Deng et al., 2019**). To generate C-cCCTomics UAS-sgRNA or UAS-Cas9 variants lines, attB vectors were injected and integrated into the attP40, attP2, or VK00005 through phiC31-mediated gene integration.

All flies generated in this research were selected by mini-white and confirmed by PCR.

## Behavioral assays

Unmated male or female flies of 4–5 d were used in circadian rhythm assays. Before measurement, flies were entrained under 12 hr light:12 hr dark cycle at 25°C for at least 3 d and then transferred to dark–dark condition for 7 d.

Virgin flies of 4–5 d were used in sleep assays. Flies were entrained to a 12 hr light:12 hr dark cycle at 25°C for 2 d to eliminate the effect of $CO_2$ anesthesia before sleep record. Sleep was defined as 5 min or longer immobility (**Hendricks et al., 2000**; **Shaw et al., 2000**) and analyzed by in-house scripts as previously described (**Dai et al., 2021**; **Dai et al., 2019**; **Deng et al., 2019**).

Locomotion was obtained as previously described (**Dai et al., 2021**). Locomotion activity was measured and analyzed by Actogram J plugin (**Dai et al., 2019**). MAI and EAI were defined as the ratio of last 3 hr activity before light-on or light-off accounts to last 6 hr activity before light-on or light-off, further subtracted by 0.5 (Index = sum(3 hr)/sum(6 hr)–0.5) (**Harrisingh et al., 2007**; **Im and Taghert, 2010**; **Seluzicki et al., 2014**) and analyzed by an in-house Python script (see Source code). Each experiment was repeated two or three times.

## Heat shock and drug treatment

For hsFLP-mediated cKO, flies of 4–6 d were heat shocked at 37°C during ZT10 to ZT12 for 4 d. They were reared at 20°C for another 4 d and then dissected.

For mifepristone (RU486)-induced cKO, flies of 4–6 d were treated with 500 µM RU486 mixed in corn food and then dissected 4 d later.

## Immunohistochemistry and confocal imaging

For all imaging without staining, adult flies were anesthetized on ice and dissected in cold phosphate buffered saline (PBS). Brains or ventral nerve codes (VNCs) were fixed in 2% paraformaldehyde (weight/volume) for 30 min, washed with washing buffer (PBS with 1% Triton X-100, v/v, 3% NaCl, g/ml) for 7 min three times, and mounted in Focusclear (Cell Explorer Labs, FC-101).

For imaging with staining, brains and VNCs were fixed for 30 min and washed for 15 min three times. Then they were blocked in PBSTS, incubated with primary antibodies, washed with washing buffer, incubated with second antibodies, and mounted as described previously (**Dai et al., 2021**; **Dai et al., 2019**).

All brains or VNCs were imaged on Zeiss LSM710 or Zeiss LSM880 confocal microscope and processed with Imaris.

The following primary antibodies were used: mouse anti-PDF (1:200, DSHB), rabbit anti-TH (1:1000, Novus Biologicals), and rabbit anti-LK (1:1000, Rao Lab, this paper). Rabbit anti-DSK (1:1000) was a gift from Dr. C. Zhou Lab (Institute of Zoology, Chinese Academy of Science) (**Wu et al., 2020**). The following secondary antibodies were used: Alexa Fluor goat anti-mouse 488 (1:1000, Invitrogen) and Alexa Flour goat anti-rabbit 488/633 (1:1000, Invitrogen).

For *Figure 2*, the number of TH-positive neurons was counted with Imaris Spots plugin.

## Quantification and statistics

MAI, MAPI, EAI, EAPI, power, and period were calculated by Python or R scripts. ZT0 was set as the time point when light was on and ZT12 was set as the time point for light off. Activity bins started at ZT0 and each was calculated as a sum of the total activity within 30 min. Flies were regarded as dead and removed if their activity value within the last two bins was 0. A representative 24 hr activity pattern was the average between the corresponding activity bins from two consecutive days. To minimize effects from singular values, each flies' activity was normalized using the following formula:

$$Nor\_b_i = \frac{b_i - \min(b_0, \ldots, b_{48})}{\max(b_0, \ldots, b_{48}) - \min(b_0, \ldots, b_{48})}$$

where $b_i$ is the activity value for a certain bin. $\min(b_0,...,b_{48})$ is the minimal bin value within 24 hr, and $\max(b_0,...,b_{48})$ is the maximal bin value within 24 hr. $Nor\_b_i$ is the final normalized bin value for a certain bin from a given fly. Normalized activity was used for the following analysis.

Morning activity arise (M_arise) was defined as the radian between the activity curve (ZT21-ZT22) and the time coordinate. Morning activity plateau (M_plateau) was defined as the radian between the activity curve (ZT22.5-ZT24) and the time coordinate. Evening activity arise (E_arise) was defined as the radian between the activity curve (ZT8-ZT11.5) and the time coordinate. Evening activity plateau (E_plateau) was defined as the radian between the activity curve (ZT10.5-ZT12) and the time coordinate. MAPI was calculated by subtracting M_arise from M_plateau, and EAPI was calculated by subtracting E_arise from E_plateau.

The original activity data from seven consecutive days in dark–dark condition was used for power and period calculation as described (*Geissmann et al., 2019*). Each fly's periodogram was calculated based on chi-square algorithm (*Sokolove and Bushell, 1978*), and flies with a null power value were regarded as arrhythmic.

All statistical analyses were carried out with Prism 8 (GraphPad software). The Kruskal–Wallis ANOVA followed by Dunn's post test was used to compare multiple columns.

## Acknowledgements

We are grateful to Xiaofei Liu, Xueying Wang, and Yile Jiao for sgRNA design and cloning; Linyi Zhang, Wenli Xu, Yuqing Gong, and Quiquan Chen for assistance with behavior assays; Xiao Dong for cloning; Xinwei Gao for assistance with imaging; Pingping Yan, Lan Wang, Yonghui Zhang, and Haixia Zeng for fly rearing; Enxing Zhou and Wei Yang for *Drosophila* activity tracing scripts; Yujie Li for cartoon illustration; Donggen Luo, Chuan Zhou, Gerald M Rubin, Vienna Drosophila RNAi Center and Bloomington Drosophila Stock Center for flies; and Yuh-Nung Jan for pACU2 construct.

## Additional information

### Funding

| Funder | Grant reference number | Author |
| --- | --- | --- |
| Chinese Institute for Brain Research, Beijing | | Yi Rao |

The funders had no role in study design, data collection and interpretation, or the decision to submit the work for publication.

### Author contributions

Renbo Mao, Conceptualization, Data curation, Formal analysis, Validation, Investigation, Visualization, Methodology, Writing – original draft, Project administration, Writing – review and editing; Jianjun Yu, Conceptualization, Data curation, Software, Formal analysis, Validation, Investigation, Visualization, Methodology; Bowen Deng, Data curation, Investigation, Found the phenotype of CNMa mutant; Xihuimin Dai, Investigation, Writing – review and editing; Yuyao Du, Sujie Du, Data curation,

Investigation; Wenxia Zhang, Resources, Supervision, Project administration; Yi Rao, Resources, Supervision, Funding acquisition, Writing – original draft, Project administration, Writing – review and editing

### Author ORCIDs
Renbo Mao ⓘ https://orcid.org/0000-0002-7310-9862
Yi Rao ⓘ http://orcid.org/0000-0002-0405-5426

Reviewer #1 (Public Review): https://doi.org/10.7554/eLife.91927.3.sa1
Reviewer #2 (Public Review): https://doi.org/10.7554/eLife.91927.3.sa2
Reviewer #3 (Public Review): https://doi.org/10.7554/eLife.91927.3.sa3
Author response https://doi.org/10.7554/eLife.91927.3.sa4

## Additional files

### Supplementary files
• Supplementary file 1. List of conditional chemoconnectome knockin flies.
• Supplementary file 2. cCCT knockin strategy cannot rescue all knockout phenotype.
• Supplementary file 3. List of sgRNAs targeting CCT genes.
• Supplementary file 4. CCT gene expression profile of clock neuron.
• Supplementary file 5. List of CCT genes intersected with Clk856 drivers.
• Supplementary file 6. Phenotypes of CCT genes knocking out in clock neurons.
• Supplementary file 7. Arrhythmicity related to *Figure 6R*.
• Supplementary file 8. Phenotypes of candidate CCT genes knockout in clock neurons.
• Supplementary file 9. Conditional knockout of VGlut in DN1s.
• Supplementary file 10. Sequence of pBSK-attB-FRT-HpaI-T2A-EGFP-FRT, PM04, pMsgNull, Cas9. M0, Cas9.M6, Cas9.M9, and UAS-sgRNA structure.
• MDAR checklist
• Source code 1. Python and R script for *Drosophila* activity analysis.

### Data availability
All data generated or analysed during this study are included in the manuscript and supporting files; source data files have been provided for Figures 1 to 7. Codes for fly activity analysis have been uploaded as Source code.

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
