## [Editor Report · eLife assessment]

This article expands the genetic toolset that was previously developed by the Rao Lab to introduce the conditional downregulation of neurotransmission components in *Drosophila*. As a proof of principle, the authors tested their new collection and provide evidence of the contribution of CNMamide (a neuropeptide) to the temporal control of locomotor activity patterns. These are overall **important** findings supported by **compelling** evidence.

---

## [Referee Report · Reviewer #1 (Public Review)]

Summary:

The paper of Mao et al. expands the genetic toolset that was previously developed by the Rao lab (Denfg et al 2019) to introduce the conditional KO or downregulation of neurotransmission components in Drosophila. The authors then use these tools to investigate neurotransmission in the the clock neurons of the *Drosophila* brain. They first test some known components and then analyze the contribution of the CNMa neuropeptide and its receptor to the circadian behavior. The results indicate that CNMA acts from a subset of DN1ps (dorsal clock neurons) to set the phase of the morning peak of locomotor activity in light:dark cycles, with an advanced morning activity in the absence of the neuropeptide. Interestingly, the receptor for the PDF neuropeptide appears to be acting in some of the CNMa neurons to control morning activity.

Strengths/weaknesses:

This is clearly a very useful new set of tools to restrict the manipulation of these components to specific neuronal populations, and overall (see specific points below), the paper is convincing to show that the tools indeed allow to efficiently and specifically eliminate neuropeptides/receptors from subsets of neurons. The analysis of the CNMa function in the clock network reveals a new and interesting function for CNMa in the control of morning anticipation in LD conditions. This function appears to depend on CNMA_expressing DN1ps.

Comment on revised version:

I believe that the authors properly addressed the main points that were raised in my comment on version 1.

---

## [Referee Report · Reviewer #2 (Public Review)]

Original Review:

In this study Mao and co-workers deliver a substantial suite of genetic tools in support of the senior author's recent proposal to create a "chemoconnectomic" tool kit for the expression mapping and conditional disruption of specific neurotransmitter systems with fly neurons of interest. Specifically, they describe the creation of two toolsets for recombination-based and CRISPR/Cas9-based conditional knockouts of genes supporting neurotransmitter and neuromodulator function and Flp-Out and Split-LexA toolkit for the examination of gene expression within defined subsets of neurons. The authors report the creation of conditional genetic tools for the disruption/mapping of approximately 200 chemoconnectomic gene products, an examination of the general effectiveness of these tools in the fly brain and apply them to the circadian clock network in an attempt to reveal new information regarding the transmitter/modulator systems involved in daily behavioral timing. The authors provide clear evidence of the effectiveness of the new methods along with a transparent assessment of the variability of the tools. In addition, they present evidence that the neuro peptide CNMa influences the morning peak of daily activity in the fly by regulating the timing of activity increases in anticipation of dawn.

A major strength of the study is the transparent assessment of the effectiveness and variability of the conditional genetic approaches developed by the authors. The authors have largely achieved their aims and the study therefore represents a major delivery on the promise of chemoconnectomics made by the senior author in 2019 (Neuron, Vol. 101, p. 876). Though there are some concerns about the variability of knockout effectiveness, off target effects of the knockout strategies, and (especially) the accuracy of the gene expression approach, the tools created for this study will almost certainly be useful for the field and support a great deal of future work.

Comments on revised version:

The authors have responded to each of my concerns. Most importantly, they have made the discrepancies within the study and between the study and previously published work clearer to the reader. they have also corrected statements that are not consistent with the current state of the field. The issue regarding opposing effects of PDF signaling and CNMa, which was also raised by Reviewer One still stands, notwithstanding the edits made to the text.

---

## [Referee Report · Reviewer #3 (Public Review)]

Summary:

Mao and colleagues generated powerful reagents to genetically analyse chemical communication (CCT) in the brain, and in the process uncovered a function for the CNMa neuropeptide expressed in a subset of DN1p neurons that contributes to the temporal organization of locomotor activity, i.e., the timing of morning anticipation.

Strengths:

The strength of the manuscript relies in the generation/characterization of new tools for conditional targeting a well-defined set of CCT genes along with the design and testing of improved versions of Cas9 for efficient knock out. Such invaluable resources will be of interest to the whole community. The authors employed these tools and intersectional genetics to provide an alternative profiling of clock neurons, which is complementary to the ones already published. Furthermore, they uncovered a role for CNMamide, expressed in two DN1ps, in the timing of morning anticipation.

Weaknesses:

All prior concerns have been addressed.

---

## [Author Response]

The following is the authors’ response to the original reviews.

We are grateful to the reviewers for their constructive comments. The following is our point-to-point responses.

**Reviewer #1 (Recommendations For The Authors):**
Point 1- Abstract: advanced morning peak « opposite » to pdf/pdfr mutants. To my knowledge, the alteration of PDF/PDFR suppresses the morning peak. I am not sure that an advance of the peak is « opposite » to its inhibition?

Mutants with disruptions in CNMa or CNMaR display advanced morning activity, indicating an enhanced state. Mutants with disruptions in Pdf or Pdfr exhibit no morning anticipation, suggesting a promoting role of these genes in morning anticipation. Therefore, our revised version is: “Specific elimination of each from clock neurons revealed that loss of the neuropeptide CNMa in two posterior dorsal clock neurons (DN1ps) or its receptor (CNMaR) caused advanced morning activity, indicating a suppressive role of CNMa-CNMaR on morning anticipation, opposite to the promoting role of PDF-PDFR on morning anticipation.” (Line 43-51)

Point 2- Fig 1K-L: the authors should show the sleep phenotype of the homozygous nAChRbeta2 mutant (if not lethal) for a direct comparison with the FRT/FLP genotype and thus evaluate the efficiency of the system.

We have incorporated sleep profiles of nAChRbeta2 mutant and W1118 into Fig 1K-L. nAChRbeta2 mutants (red) exhibited a sleep amount comparable to that of pan-neural nAChRbeta2 knockout flies (dark red), as shown below.

**Author response image 1. sa4fig1:** 

Point 3- Dh31-EGFP-FRT expression patterns look different in figS1 A (or fig1 H) and J. why that?

We re-examined the original data. Both (with R57C10-GAL4 for Fig. S1A, right, S1J, left) are Dh31EGFP.FRT samples displayed below which demonstrated consistent primary expression subsets. Any observed disparities in region "e" could potentially be attributed to variations during dissection.

**Author response image 2. sa4fig2:** 

Point 4- The knockdown experiments with the elav-switch (RU486) system (fig S2) do not seem to be as efficient as the HS-FLP system (fig 1H-J). The conclusions on the efficiency should be toned down.

We have revised accordingly: "Near Complete Disruption of Target Genes by GFPi and Flp-out Based cCCTomics" (Line 130): "Knocking out at the adult stage using either hsFLP driven Flp-out (Golic and Lindquist, 1989) (Fig. 1H-1J) or neural (elav-Switch) driven shRNAGFP (Nicholson et al., 2008; Osterwalder et al., 2001) (Fig. S2A-S2I), also resulted in the elimination of most, though not all, GFP signals." (Line 145-149)

Point 5- Fig 2H-J: the LD behavioral phenotype of pdfr pan-neuronal cripsr does not seem to correspond to what is described in the literature for the pdfr mutant (han), see hyun et al 2005 (no morning anticipation and advanced evening peak). I understand that the activity index is lower than controls but fig2H shows a large anticipatory activity that seems really unusual, and no advanced evening peak is observed. I think that the authors should show the CRISPR flies and pdfr mutants together, to better compare the phenotypes.

Thank you for pointing out that the phenotypes of pan-neuronal knockout of PDFR by unmodified Cas9 (Fig. 2H-2I of the previous version) whose morning anticipation still exist (Fig, 2H of the previous manuscript), although the significant decrease of morning anticipation index (Fig 2I of the previous manuscript) and advanced evening activity are not as pronounced as observed in han5304 (Fig. 3C in Hyun et al., 2005).

First, we have separated the activity plots of Fig. 2H of previous manuscript, as shown below. The activity from ZT18 to ZT24 shows a tendency of decreasing from ZT18 to ZT21 and a tendency of increasing from ZT21 to ZT24. The lowest activity before dawn during ZT18 to ZT24 shows at about ZT21, and the activity at ZT18 is comparable to the activity at ZT24. This is significantly different compared to the two control groups, whose activity tends to increase activity from ZT18 to ZT24 with an activity peak at ZT24.

The activity from ZT6 to ZT12 increased much faster in Pdfr knockout flies and get to an activity plateau at about ZT11 compared to two control groups with a slower activity increasing from ZT6 to ZT12 with no activity plateau but an activity peak at ZT12.

**Author response image 3. sa4fig3:** 

Second, we have incorporated the phenotype of Pdfr mutants we previously generated (Pdfr-attpKO Deng et al., 2019) with Pdfr pan-neuronal knockout by Cas9.HC. This mutant lacks all seven transmembrane regions of Pdfr (a). The phenotypes are very similar between Pdfr-attpKO flies and Pdfr pan-neuronal knockout flies. In this experimental repeat, we found that a much more obvious advanced evening activity peak is observed both in pan-neuronal knockout flies and Pdfr-attpKO flies.

To further analyze the phenotypes of Pdfr pan-neuronal knockout flies by Cas9.HC, we referred to the literature. The activity pattern at ZT18 to ZT24 (activity tends to decrease from ZT18 to ZT21 and tends to increase from ZT21 to ZT24, with the lowest activity before dawn occurring at about ZT21, and activity at ZT18 comparable to activity at ZT24) is also reported in Pdfr knockout flies such as Fig3C and 3H in Hyun et al., 2005, Fig 2B in Lear et al., 2009, Fig 3B in Zhang et al., 2010, Fig .5A in Guo et al., 2014, and Fig 5B in Goda et al., 2019. Additionally, the less pronounced advanced evening activity peak compared to han5304 (Fig. 3C in Hyun et al., 2005) is also reported in Fig. 2B in Lear et al., 2009, Fig. 3B in Zhang et al., 2010, and Fig. 5B in Goda et al., 2019. We consider that this difference is more likely to be caused by environmental conditions or recording strategies (DAM system vs. video tracing).

Therefore, we revised the text to: “Pan-neuronal knockout of Pdfr resulted in a tendency towards advanced evening activity and weaker morning anticipation compared to control flies (Fig. 2H-2I), which is similar to Pdfr-attpKO flies. These phenotypes were not as pronounced as those reported previously, when han5304 mutants exhibited a more obvious advanced evening peak and no morning anticipation (Hyun et al., 2005)”.

**Author response image 4. sa4fig4:** 

Point 6-The authors should provide more information about the DD behavior (power is low, but how about the period of rhythmic flies, which is shortened in pdf (renn et al) and pdfr (hyun et al) mutants).

We have incorporated period data into Fig. 2I. Indeed, conditional knock out of Pdfr by Cas9.HC driven by R57C10-GAL4 shortens the period length, as shown below (previous data), also in Fig. 2I of the revised version.

In the revised Fig. 2I, we tested 45 Pdfr-attpKO flies during DD condition (3 out of 48 flies died during video tracing in DD condition), and only one fly was rhythmic. In contrast, 9 out of 48 Pdfr pan-neuronal knockout flies were rhythmic.

**Author response image 5. sa4fig5:** 

Point 7- P15 and fig6. The authors indicate that type II CNMa neurons do not show advanced morning activity as type I do, but Figs 6 I and K seem to show some advance although less important than type I. I am not sure that this supports the claim that type I is the main subset for the control of morning activity. This should be toned down.

We have re-organized Fig. 6 and revised the summary of these results as: “However, Type II neurons-specific CNMa knockout (CNMa ∩ GMR91F02) showed weaker advanced morning activity without advanced morning peak (Fig. 6N), while Type I neurons-specific CNMa knockout did (Fig. 6J), indicating a possibility that these two type I CNMa neurons constitute the main functional subset regulating the morning anticipation activity of fruit fly”. (Line 400-405)

Point 8- Figs 6M and N: is power determined from DD data? if yes, how about the period and arrhythmicity? Please also provide the LD activity profiles for the mutants and rescued pdfr genotypes.

Yes, the power was determined from the DD data. In the new version of the manuscript, we have included the activity plots for the LD phase in supplementary Fig S13, as well as shown below (A, B), and the period and arrhythmicity data for the DD phase in Fig. 6S and Table S7. We have also refined the related description as follows: “Moreover, knocking out Pdfr by GMR51H05, GMR79A11 and CNMa GAL4, which cover type I CNMa neurons, decreased morning anticipation of flies (Fig. 6T, Fig. S13B). However, the decrease in morning anticipation observed in the Pdfr knockout by CNMa-GAL4 was not as pronounced as with the other two drivers. Because the presumptive main subset of functional CNMa is also PDFR-positive, there is a possibility that CNMa secretion is regulated by PDF/PDFR signal”. (Line 413-419)

**Author response image 6. sa4fig6:** 

Point 9- Fig 7: does CNMaR affect DD behavior? This should be tested.

We analyzed the CNMaR-/- activity in the dark-dark condition over a span of six days. Results revealed a higher power in CNMaR mutants compared to control flies (Power: 93.5±41.9 (CNMaR-/-, n=48) vs 47.3±31.6 (w1118, n=47); Period: 23.7±0.3 h (CNMaR-/-, n=46) vs 23.7±0.3 h (w1118, n=47); arrhythmic rate 2/48 (CNMaR-/-) vs 0/47 (w1118)). Considering that mutating CNMa had no obvious effect on DD behavior, even if CNMaR affects DD behavior, it cannot be attributed to CNMa signal, we did not further repeat and analyze DD behavior of CNMaR mutant. We believe this raises another question beyond the scope of our current discussion.

**Reviewer #2 (Recommendations For The Authors):**
Point 1-One major concern is the apparent discrepancies in clock network gene expression using the Flp-Out and split-LexA approaches compared to what is known about the expression of several transmitter and peptide-related genes. For example, it is well established that the 5th-sLNv expresses CHAT (along with a single LNd), yet there appears to be no choline acetyltransferase (ChAT) signal in the 5th-sLNv as assayed by the split-LexA approach (Fig. 4). This approach also suggests that DH31 is expressed in the s-LNvs, which, as one of the most intensely studied clock neuron are known to express PDF and sNPF, but not DH31. The results also suggest that the sLNvs express ChAT, which they do not. Remarkably PDF is not included in the expression analysis, this peptide is well known to be expressed in only two subgroups of clock neurons, and would therefore be an excellent test case for the expression analysis in Fig. 4. PDF should therefore be added to analysis shown in Fig. 4. Another discrepancy is PdfR, which split LexA suggests is expressed in the Large LNvs but not the small LNvs, the opposite of what has been shown using both reporter expression and physiology. The authors do acknowledge that discrepancies exist between their data and previous work on expression within the clock network (lines 237 and 238). However, the extent of these discrepancies is not made clear and calls into question the accuracy of Flp-Out and Split LexA approaches.

The concerns mentioned above are:

(1) sLNvs express PDF and sNPF but not Dh31;

(2) ChAT presents in 5th-sLNv and one LNd but not in other sLNvs;

(3) PDFR presents in sLNvs but not l-LNvs.

(4) PDF is not included in the analysis.

To verify the accuracy of these intersection analyses, all related to PDF positive neurons (except 5th-sLNv and LNds), we stained PDF and examined the co-localization between PDF-positive LNvs and the respective drivers ChAT-KI-LexA, Pdfr-KI -LexA, Dh31-KI -LexA, and Pdf-KI -LexA.

First, Dh31-KI-LexA labeled four s-LNvs, as shown below (also in Fig. S9A). Therefore, the results of the intersection analysis of Dh31-KI-LexA with Clk856-GAL4 are correct. The difference in the results compared to previous literature is attributed to Dh31-KI-LexA labels different neurons than the previous driver or antibody.

Second, no s-LNv was labeled by ChAT-KI -LexA as shown below. We rechecked our intersection data and found that we analyzed 10 brains of ChAT-KI-LexA∩Clk856-GAL4 while only two brains showed sLNvs positively. To enhance the accuracy of intersection analysis results, we marked all positive signal records when positive subsets were found in less than 1/3 of the total analyzed brains (Table S4).

Third, one l-LNv and at least two s-LNvs were labeled by Pdfr-KI-LexA, as shown below (also in Fig. S9B). Fourth, Pdf-KI-LexA labels all PDF-positive neurons, but the intersection analysis by Pdf-KI-LexA and Clk856-GAL4 only showed scattered signals, as shown below (D, also in Fig. S9C). For these cases, we found some positive signals expected but not observed in our dissection. The possible reason could be the inefficiency of LexAop-FRT-myr::GFP driven by LexA. Therefore, our intersection results must miss some positive signals.

**Author response image 7. sa4fig7:** 

Finally, we revised the text to (Line 286-317):

To assess the accuracy of expression profiles using CCT drivers, we compared our dissection results with previous reports. Initially, we confirmed the expression of CCHa1 in two DN1s (Fujiwara et al., 2018), sNFP in four s-LNvs and two LNds(Johard et al., 2009), and Trissin in two LNds (Ma et al., 2021), aligning with previous findings. Additionally, we identified the expression of nAChRα1, nAChRα2, nAChRβ2, GABA-B-R2, CCHa1-R, and Dh31-R in all or subsets of LNvs, consistent with suggestions from studies using ligands or agonists in LNvs (Duhart et al., 2020; Fujiwara et al., 2018; Lelito and Shafer, 2012; Shafer et al., 2008) (Table S4).

Regarding previously reported Nplp1 in two DN1as (Shafer et al., 2006), we found approximately five DN1s positive for Nplp-KI-LexA, indicating a broader expression than previously reported. A similar pattern emerged in our analysis of Dh31-KI-LexA, where four DN1s, four s-LNvs, and two LNds were identified, contrasting with the two DN1s found in immunocytochemical analysis (Goda et al., 2016). Colocalization analysis of Dh31-KI-LexA and anti-PDF revealed labeling of all PDF-positive s-LNvs but not l-LNvs (Fig S9A), suggesting that the differences may arise from the broader labeling of 3' end knock-in LexA drivers or the amplitude effect of the binary expression system. The low protein levels might go undetected in immunocytochemical analysis. This aligns with transcriptome analysis findings showing Nplp1 positive in DN1as, a cluster of CNMa-positive DN1ps, and a cluster of DN3s (Ma et al., 2021), which is more consistent with our dissection.

Despite the well-known expression of PDF in LNvs and PDFR in s-LNvs (Renn et al., 1999; Shafer et al., 2008), we did not observe stable positive signals for both in Flp-out intersection experiments, although both Pdf-KI-LexA and Pdfr-KI-LexA label LNvs as expected (Fig S9B-S9C). We also noted fewer positive neurons in certain clock neuron subsets compared to previous reports, such as NPF in three LNds and some LNvs (Erion et al., 2016; He et al., 2013; Hermann et al., 2012; Johard et al., 2009; Lee et al., 2006) and ChAT in four LNds and the 5th s-LNv (Johard et al., 2009; Duhart et al., 2020) (Table S4). We attribute this limitation to the inefficiency of LexAop-FRT-myr::GFP driven by LexA, acknowledging that our intersection results may miss some positive signals.

Point 2-Related to this, the authors rather inaccurately suggest that the field's understanding of PdfR expression within the clock neuron network is "inconsistent" and "variable" (lines 368-377). This is not accurate. It is true that the first attempts to map PdfR expression with antisera and GAL4s were inaccurate. However, subsequent work by several groups has produced strong convergent evidence that with the exception of the l-LNvs after several days post-eclosion, PdfR is expressed in the Cryptochrome expressing a subset of the clock neuron network. This section of the study should be revised.

We thank the reviewer for pointing this out. As we have already addressed and revised the related part in the RESULTS section (Line 308-317), we have now removed this part from the DISCUSSION section of the revised version.

Point 3-One minor issue that would avoid unnecessary confusion by readers familiar with the circadian literature is the say that activity profiles are plotted in the study. The authors have centered their averaged activity profiles on the 12h of darkness. This is the opposite of the practice of the field, and it leads to some initial confusion in the examination of the morning and evening peak data. The authors may wish to avoid this by centering their activity plots on the 12h light phase, which would put the morning peak on the left and the evening peak on the right. This is the way the field is accustomed to examining locomotor activity profiles.

The centering of averaged activity profiles on the 12 h of darkness is done to highlight the phenotype of advanced morning activity. To prevent any confusion among readers, we have included a sentence in the figure legend explaining the difference in our activity profiles compared to previous literatures: "Activity profiles were centered of the 12 h darkness in all figures with evening activity on the left and morning activity on the right, which is different from general circadian literatures. (Fig. 2H legend)" (Line 957-959)

Point 4-The authors conclude that the loss of PDF and CNMa have opposite effects on the morning peak of locomotor activity (line 392). But they also acknowledge, briefly, that things are not that simple: loss of CNMa causes a phase advance, but loss of PDF causes a loss or reduction in the anticipatory peak. It is still significant to find a peptide transmitter with the clock neuron network that regulates morning activity, but the authors should revise their conclusion regarding the opposing actions of PDF and CNMa, which is not well supported by the data.

We have revised the relevant parts.

ABSTRACT: “Specific elimination of each from clock neurons revealed that loss of the neuropeptide CNMa in two posterior dorsal clock neurons (DN1ps) or its receptor (CNMaR) caused advanced morning activity, indicating a suppressive role of CNMa-CNMaR on morning anticipation, opposite to the promoting role of PDF-PDFR on morning anticipation.” (Line 43-48)

DISCUSSION: “Furthermore, given that the morning anticipation vanishing phenotype of Pdf or Pdfr mutant indicates a promoting role of PDF-PDFR signal, while the enhanced morning anticipation phenotype of CNMa mutant suggests an inhibiting role of CNMa signal, we consider the two signals to be antagonistic.” (Line 492-495)

Point 5-The authors should acknowledge, cite, and incorporate the substantive discussion of CNMa peptide and the DN1p neuronal class in Reinhard et al. 2022 (Front Physiol. 13: 886432).

We have revised the text accordingly and cited this paper: “Type I with two neurons whose branches projecting to the anterior region, as in CNMa∩GMR51H05, CNMa∩Pdfr, and CNMa∩GMR79A11 (Fig. 6E, 5G, 6H), and type II with four neurons branching on the posterior side with few projections to the anterior region, as in CNMa∩GMR91F02 (Fig. 6F). These two types of DN1ps’ subsets were also reported and profound discussed previously (Lamaze et al., 2018; Reinhard et al., 2022)”. (Line 393-397)

**Reviewer #3 (Recommendations For The Authors):**
Point 1-Throughout the manuscript figure legends (axis, genotypes, etc) are too small to be appreciated. Fig. 1. Panel A. The labels are very difficult to read.

We have attempted to enlarge the font as much as possible in the revised version.

Point 2-Fig. 1. H-J Why is efficiency not mentioned in all the examples?

In the revised manuscript, the results of Fig 1H-1J are discussed in the revised version (Line 145-147). The reason that we did not calculate the exact efficiency is that the GFP intensity is not stable enough which might change during dissection, mounting or intensity of laser in our experimental process. Therefore, in all results related to GFP signal (Fig. 1B-1J, Fig. S1, Fig. S2, Fig. 2B-2D), we relied on qualitative judgment rather than quantitative judgment, unless the GFP signal was easily quantifiable (such as in cases with limited cells or no GFP signal in the experimental group).

Point 3-Fig. 1. Panel L, left (light phase): the statistical comparisons are not clearly indicated (the same happens in Figs 3Q and 3R).

We have now re-arranged Fig. 1L and Fig. 3Q-3R to make the statistical comparisons clear in the new version.

Point 4-Line 792. Could induced be introduced?

Yes, we have now corrected this typo.

Point 5-Fig. S1. Check labels for consistency. GMR57C10 Gal4 driver is most likely R57C10.

We have now revised the labels (Fig. S1).

Point 6-Fig. S2. If the experiments were repeated and several brains were observed, the authors should include the efficiency and the number of flies as reported in Fig. S1.

We have now added the number of flies in Fig. S2 as reported in Fig. S1. As Response to Point 2 mentioned, due to the instability of the GFP signal, we are unable to provide a quantitative efficiency in this context.

Point 7-Fig S4. The fig legend describes panels I-J which are not shown in the current version of the manuscript.

We now have deleted them.

Point 8-Fig 2I. Surprising values for morning anticipation indexes even for controls 0.5 would indicate ¨no anticipation¨; in controls, the expected values would be >>0.5, as most of the activity is concentrated right before the transition. Could the authors explain this unexpected result?

We have revised the description of the calculation in the methods section (Line 612). After calculating the ratio of the last three hours of activity to the total six hours of activity, the results were further subtracted by 0.5. Therefore, the index should be ≤0.5. When the index is equal to 0, it indicates no morning anticipation.

Point 9-Fig 2K/L. The authors mention that not all genes are effectively knocked out with their strategy. Could this be accounted for the specific KD strategy, its duration, or the promotor strength? It is surprising no explanation is provided in the text (page 9 line 179).

In our pursuit of establishing a broadly effective method for gene editing, Fig. 2H-2L and Fig. 2D revealed that previous attempts have fallen short of achieving this objective. The observed inefficiency may be attributed to the intensity of the promoter, resulting in inadequate expression. Alternatively, the insufficient duration of the operation may also contribute to the lack of success. However, in the context of sleep and rhythm research applications, the age of the fruit fly tests is typically fixed, limiting the potential to enhance efficiency by extending the manipulation time. Moreover, increasing the expression level may pose challenges related to cytotoxicity, as reported in previous studies (Port et al., 2014). We refrain from offering specific explanations, as we lack a definitive plan and cannot provide additional robust evidence to support the above speculations. Consequently, in our ongoing efforts, we aim to enhance the efficiency of the tool system while operating within the current constraints.

Point 10-Page 9, line 179. Can the authors include a brief description of the reason for the different modifications? Only one was referenced.

We have revised related part in the manuscript (Line 223-231):

Cas9.M9: We fused a chromatin-modulating peptide (Ding et al., 2019), HMGN1 183 (High mobility group nucleosome binding domain 1), at the N-terminus of Cas9 and HMGB1 184 (High mobility group protein B1) at its C-terminus with GGSGP linker, termed Cas9.M9.

Cas9.M6: We also obtained a modified Cas9.M6 with HMGN1 at the N-terminus and an undefined peptide (UDP) at the C-terminus. (NOTE：UDP was gained by accident)

Cas9.M0: We replaced the STARD linker between Cas9 and NLS in Cas9.HC with GGSGP the linker (Zhao et al., 2016), termed Cas9.M0

Point 11-The authors tested the impact of KO nAChRβ2 across the different versions of conditional disruption (Fig 1K-L, Fig 2L, Fig 3R). It is surprising they observe a difference in daytime sleep upon knocking down with Cas9.HC (2L) but not with Cas9.M9 (3R) and the reverse is seen for night-time sleep. Could the authors provide an explanation? Efficiency is not the issue at stake, is it?

In Fig. 2K, the day sleep of flies (R57C10-GAL4/UAS-sgRNAnAChRbeta2; UAS-Cas9/+) was significantly decreased compared to flies (R57C10-GAL4/UAS-sgRNAnAChRbeta2; +/+), but not when compared to flies (R57C10-GAL4/+; UAS-Cas9/+). Our criterion for asserting a difference is that the experimental group must show a significant distinction from both control groups. Therefore, we concluded that there was no significant difference between the experimental group and the control groups in Fig. 2K.

Point 12-Fig. 4. Which of the two strategies described in A-B was employed to assemble the expression profile of CCT genes in clock neurons shown in C? This information should be part of the fig legend.

We have now revised the legend as follows: “(A-B) Schematic of intersection strategies used in Clk856 labelled clock neurons dissection, Flp-out strategy (A) and split-LexA strategy (B). The exact strategy used for each gene is annotated in Table S5.”

Point 13-Similarly, how many brains were analyzed to give rise to the table shown in C?

We have now revised the legend of Table S4 to address this concern. As indicated in: “The largest N# for each gene in Table S4 is the brain number analyzed for each gene”.

Point 14-Finally, the sentence ¨The figure is...¨ requires revision.

We have now revised it: “The exact cell number for each subset is annotated in Table S4”.

Point 15-Legend to Table S3. The authors have done an incredible job testing many gRNAs for each gene potentially relevant for communication. However, there is very little information to make the most out of it; for instance, the legend does not inform why many of the targeted genes do not appear to have been tested any further. It would be useful to the reader to discern whether despite being the 3 most efficient gRNAs, they were still not effective in targeting the gene of interest, or whether they showed off-targets, or it was simply a matter of testing the educated guesses. This information would be invaluable for the reader.

First, we designed and generated transgenic UAS-sgRNA fly lines for all these sgRNAs. We randomly selected 14 receptor genes, known for their difficulty in editing based on our experience, to assess the efficiency of our strategy, as depicted in Fig. 3M-3P, Fig. S5, and Fig. S6. We believe these results are representative and indicative of the efficiency of sgRNAs designed using our process and applied with the modified Cas9.

Secondly, we acknowledge your valid concern. While we selected sgRNAs with no predicted off-target effects through various prediction models (outlined in the Methods under C-cCCTomics sgRNA design), we did not conduct whole-genome sequencing. Consequently, we can only assert that the off-target possibility is relatively low. To address potential misleading effects arising from off-target concerns, it is essential to validate these results through mutants, RNAi, or alternative UAS-sgRNAs targeting the same gene.

Point 16-Table S4. Some of the data presented derives from observations made in 1-2 brains for a specific cluster; isn´t it too little to base a decision on whether a certain gene is (or not) expressed? It is surprising since the same CCT line was observed/analysed in more brains for other clusters. Can the authors explain the rationale?

The N# number represents the GFP positive number, and we have revised the legend of Table S4. The largest N# number denotes the total number of brains analyzed for a specific CCT line. It's possible that, due to variations in our dissection or mounting process, some clusters were only observed in 1-2 brains out of the total brains analyzed. To enhance the accuracy of intersection analysis results, we marked all positive signal records when positive subsets were found in less than 1/3 of the total analyzed brains (Table S4).

Point 17-The paragraph describing this data in the results section needs revising (lines 233-243).

We have now revised this. (Line 286-317)

Point 18-While it is customary for authors to attempt to improve the description of the activity patterns by introducing new parameters (i.e. MAPI and EAPI, lines 253-258) it would be interesting to understand the difference between the proposed method and the one already in use which compares the same parameter, i.e., the slope (defined as ¨the slope of the best-fitting linear regression line over a period of 6 h prior to the transition¨, i.e., Lamaze et al. 2020 and many others). Is there a need to introduce yet another one?

This approach is necessary. The slope defined by Lamaze et al. utilizes data from only 2 time points, which may not accurately capture the pattern within a period before light on or off. Linear regression is not well-suited for a single fly due to the high variability in activity at each time point, making it challenging to fit the model at the individual level. The parameters we have introduced (MAPI and EAPI) in this paper are concise and can be applied at the individual level, effectively reflecting the morning or evening anticipation characteristics of each fly.

As an alternative, the activity plot of a certain fly line could be represented by an average of all flies' activity in one experiment. This would make linear regression easier to fit. However, several independent experiments are required for statistical robustness, necessitating the inclusion of hundreds of flies for each strain in a single analysis.

Point 19-In general, the legends of supplementary figures are a bit too brief. S7 and S8: it is not clear which of the two intersectional strategies were used (it would benefit whoever is interested in replicating the experiments). Legend to Fig S8 should read ¨similar to Fig S7¨.

We have now revised the legend and included “The exact strategy used for each gene is annotated in Table S5” in the legend.

Point 20-The legend in Table S6 should clearly state the genotypes examined. What does the marking in bold refer to?

We have now revised annotation of Table S6. Marking in bold refer to results out of one SD compared to control group.

Point 21-Line 314. The sentence needs revision.

We have revised these sentences.

Point 22-Line 391 (and also in the results section). The authors attempt to describe the CNMa phenotype as the opposite of pdf/pdfr mutant phenotypes. However, no morning anticipation/advanced morning anticipation are not necessarily opposite phenotypes.

We have revised related description.

ABSTRACT: “Specific elimination of each from clock neurons revealed that loss of the neuropeptide CNMa in two posterior dorsal clock neurons (DN1ps) or its receptor (CNMaR) caused advanced morning activity, indicating a suppressive role of CNMa-CNMaR on morning anticipation, opposite to the promoting role of PDF-PDFR on morning anticipation.” (Line 43-48)

DISCUSSION: “Furthermore, given that the morning anticipation vanishing phenotype of Pdf or Pdfr mutant indicates a promoting role of PDF-PDFR signal, while the enhanced morning anticipation phenotype of CNMa mutant suggests an inhibiting role of CNMa signal, we consider the two signals to be antagonistic.” (Line 492-495)

Reference

Deng, B., Li, Q., Liu, X., Cao, Y., Li, B., Qian, Y., Xu, R., Mao, R., Zhou, E., Zhang, W., et al. (2019). Chemoconnectomics: mapping chemical transmission in *Drosophila*. Neuron 101, 876-893.e874.

Ding, X., Seebeck, T., Feng, Y., Jiang, Y., Davis, G.D., and Chen, F. (2019). Improving CRISPR-Cas9 genome editing efficiency by fusion with chromatin-modulating peptides. Crispr j 2, 51-63.

Duhart, J.M., Herrero, A., de la Cruz, G., Ispizua, J.I., Pírez, N., and Ceriani, M.F. (2020). Circadian Structural Plasticity Drives Remodeling of E Cell Output. Curr Biol 30, 5040-5048.e5045.

Erion, R., King, A.N., Wu, G., Hogenesch, J.B., and Sehgal, A. (2016). Neural clocks and Neuropeptide F/Y regulate circadian gene expression in a peripheral metabolic tissue. eLife 5, e13552.

Fujiwara, Y., Hermann-Luibl, C., Katsura, M., Sekiguchi, M., Ida, T., Helfrich-Förster, C., and Yoshii, T. (2018). The CCHamide1 neuropeptide expressed in the anterior dorsal neuron 1 conveys a circadian signal to the ventral lateral neurons in *Drosophila melanogaster*. Front Physiol 9, 1276.

Goda, T., Tang, X., Umezaki, Y., Chu, M.L., Kunst, M., Nitabach, M.N.N., and Hamada, F.N. (2016). *Drosophila* DH31 neuropeptide and PDF receptor regulate night-onset temperature preference. J Neurosci 36, 11739-11754.

Goda, T., Umezaki, Y., Alwattari, F., Seo, H.W., and Hamada, F.N. (2019). Neuropeptides PDF and DH31 hierarchically regulate free-running rhythmicity in *Drosophila* circadian locomotor activity. Sci Rep 9, 838.

Guo, F., Cerullo, I., Chen, X., and Rosbash, M. (2014). PDF neuron firing phase-shifts key circadian activity neurons in *Drosophila*. Elife 3.

He, C., Cong, X., Zhang, R., Wu, D., An, C., and Zhao, Z. (2013). Regulation of circadian locomotor rhythm by neuropeptide Y-like system in *Drosophila melanogaster*. Insect Mol Biol 22, 376-388.

Hermann, C., Yoshii, T., Dusik, V., and Helfrich-Förster, C. (2012). Neuropeptide F immunoreactive clock neurons modify evening locomotor activity and free-running period in *Drosophila melanogaster*. J Comp Neurol 520, 970-987.

Hyun, S., Lee, Y., Hong, S.T., Bang, S., Paik, D., Kang, J., Shin, J., Lee, J., Jeon, K., Hwang, S., et al. (2005). *Drosophila* GPCR Han is a receptor for the circadian clock neuropeptide PDF. Neuron 48, 267-278.

Johard, H.A., Yoishii, T., Dircksen, H., Cusumano, P., Rouyer, F., Helfrich-Förster, C., and Nässel, D.R. (2009). Peptidergic clock neurons in *Drosophila*: ion transport peptide and short neuropeptide F in subsets of dorsal and ventral lateral neurons. J Comp Neurol 516, 59-73.

Lamaze, A., Krätschmer, P., Chen, K.F., Lowe, S., and Jepson, J.E.C. (2018). A Wake-Promoting Circadian Output Circuit in *Drosophila*. Curr Biol 28, 3098-3105.e3093.

Lear, B.C., Zhang, L., and Allada, R. (2009). The neuropeptide PDF acts directly on evening pacemaker neurons to regulate multiple features of circadian behavior. PLoS Biol 7, e1000154.

Lee, G., Bahn, J.H., and Park, J.H. (2006). Sex- and clock-controlled expression of the neuropeptide F gene in *Drosophila*. 103, 12580-12585.

Lelito, K.R., and Shafer, O.T. (2012). Reciprocal cholinergic and GABAergic modulation of the small ventrolateral pacemaker neurons of *Drosophila*'s circadian clock neuron network. J Neurophysiol 107, 2096-2108.

Ma, D., Przybylski, D., Abruzzi, K.C., Schlichting, M., Li, Q., Long, X., and Rosbash, M. (2021). A transcriptomic taxonomy of *Drosophila* circadian neurons around the clock. Elife 10.

Port, F., Chen, H.M., Lee, T., and Bullock, S.L. (2014). Optimized CRISPR/Cas tools for efficient germline and somatic genome engineering in *Drosophila*. Proc Natl Acad Sci USA 111, E2967-2976.

Reinhard, N., Schubert, F.K., Bertolini, E., Hagedorn, N., Manoli, G., Sekiguchi, M., Yoshii, T., Rieger, D., and Helfrich-Förster, C. (2022). The Neuronal Circuit of the Dorsal Circadian Clock Neurons in *Drosophila melanogaster*. Front Physiol 13, 886432.

Renn, S.C., Park, J.H., Rosbash, M., Hall, J.C., and Taghert, P.H. (1999). A pdf neuropeptide gene mutation and ablation of PDF neurons each cause severe abnormalities of behavioral circadian rhythms in *Drosophila*. Cell 99, 791-802.

Shafer, O.T., Helfrich-Förster, C., Renn, S.C., and Taghert, P.H. (2006). Reevaluation of *Drosophila melanogaster*'s neuronal circadian pacemakers reveals new neuronal classes. J Comp Neurol 498, 180-193.

Shafer, O.T., Kim, D.J., Dunbar-Yaffe, R., Nikolaev, V.O., Lohse, M.J., and Taghert, P.H. (2008). Widespread receptivity to neuropeptide PDF throughout the neuronal circadian clock network of *Drosophila* revealed by real-time cyclic AMP imaging. Neuron 58, 223-237.

Zhang, L., Chung, B.Y., Lear, B.C., Kilman, V.L., Liu, Y., Mahesh, G., Meissner, R.A., Hardin, P.E., and Allada, R. (2010). DN1(p) circadian neurons coordinate acute light and PDF inputs to produce robust daily behavior in *Drosophila*. Curr Biol 20, 591-599.

Zhao, P., Zhang, Z., Lv, X., Zhao, X., Suehiro, Y., Jiang, Y., Wang, X., Mitani, S., Gong, H., and Xue, D. (2016). One-step homozygosity in precise gene editing by an improved CRISPR/Cas9 system. Cell Res 26, 633-636.